# Discovery and characterization of a specific inhibitor of serine-threonine kinase cyclin-dependent kinase-like 5 (CDKL5) demonstrates role in hippocampal CA1 physiology

Anna Castano[1†], Margaux Silvestre[2†], Carrow I Wells[3], Jennifer L Sanderson[1], Carla A Ferrer[3], Han Wee Ong[3], Yi Lang[3], William Richardson[4], Josie A Silvaroli[5], Frances M Bashore[3], Jeffery L Smith[3], Isabelle M Genereux[3], Kelvin Dempster[2], David H Drewry[3,6], Navlot S Pabla[5], Alex N Bullock[4], Tim A Benke[7*‡], Sila K Ultanir[2*‡], Alison D Axtman[3*‡]

[1]Department of Pharmacology, University of Colorado School of Medicine, Aurora, United States; [2]Kinases and Brain Development Laboratory, The Francis Crick Institute, London, United Kingdom; [3]Structural Genomics Consortium, UNC Eshelman School of Pharmacy, University of North Carolina at Chapel Hill, Chapel Hill, United States; [4]Centre for Medicines Discovery, Nuffield Department of Medicine, University of Oxford, Oxford, United Kingdom; [5]Division of Pharmaceutics and Pharmacology, College of Pharmacy and Comprehensive Cancer Center, The Ohio State University, Columbus, United States; [6]Lineberger Comprehensive Cancer Center, School of Medicine, University of North Carolina at Chapel Hill, Chapel Hill, United States; [7]Departments of Pediatrics, Pharmacology, Neurology and Otolaryngology, University of Colorado School of Medicine, Aurora, United States

*For correspondence:
tim.benke@cuanschutz.edu (TAB);
sila.ultanir@crick.ac.uk (SKU);
Alison.Axtman@unc.edu (ADA)

[†]These authors contributed equally to this work

[‡]Co-senior authors

**Abstract** Pathological loss-of-function mutations in cyclin-dependent kinase-like 5 (*CDKL5*) cause CDKL5 deficiency disorder (CDD), a rare and severe neurodevelopmental disorder associated with severe and medically refractory early-life epilepsy, motor, cognitive, visual, and autonomic disturbances in the absence of any structural brain pathology. Analysis of genetic variants in CDD has indicated that CDKL5 kinase function is central to disease pathology. *CDKL5* encodes a serine-threonine kinase with significant homology to GSK3β, which has also been linked to synaptic function. Further, *Cdkl5* knock-out rodents have increased GSK3β activity and often increased long-term potentiation (LTP). Thus, development of a specific CDKL5 inhibitor must be careful to exclude cross-talk with GSK3β activity. We synthesized and characterized specific, high-affinity inhibitors of CDKL5 that do not have detectable activity for GSK3β. These compounds are very soluble in water but blood–brain barrier penetration is low. In rat hippocampal brain slices, acute inhibition of CDKL5 selectively reduces postsynaptic function of AMPA-type glutamate receptors in a dose-dependent manner. Acute inhibition of CDKL5 reduces hippocampal LTP. These studies provide new tools and insights into the role of CDKL5 as a newly appreciated key kinase necessary for synaptic plasticity. Comparisons to rodent knock-out studies suggest that compensatory changes have limited the understanding of the roles of CDKL5 in synaptic physiology, plasticity, and human neuropathology.

## Editor's evaluation

This important study reports selective CDKL5 inhibitors that may be of high interest to investigate the role of this kinase in disease (particularly, in CDKL5 deficiency disorder) and to address unsolved issues of inconsistency in the phenotypic characterization of CDKL5-deficient knockout mice. The evidence supporting the conclusions is convincing, with rigorous biochemical, in vitro and ex vivo assays. The work will be of interest to cell and medical biologists and epileptologists working in the fields of epilepsy and neural excitation.

## Introduction

Pathological loss-of-function mutations in cyclin-dependent kinase-like 5 (*CDKL5*) (*Weaving et al., 2004*; *Bahi-Buisson et al., 2012*) cause CDKL5 deficiency disorder (CDD, OMIM 300203, 300672), a rare (incidence 1:40,000–60,000; *Lindy et al., 2018*; *Kothur et al., 2018*; *Symonds, 2017*) neuro-developmental disorder associated with severe early-life epilepsy, motor, cognitive, visual, and autonomic disturbances (*Bahi-Buisson et al., 2012*; *Bahi-Buisson et al., 2008*; *Nemos et al., 2009*; *Castrén et al., 2011*; *Melani et al., 2011*; *Fehr et al., 2015*). Current experience demonstrates that epilepsy in CDD is very medically refractory (*Müller et al., 2016*; *Demarest et al., 2019*; *Olson et al., 2019*; *Leonard et al., 2022*; with rare exception [*Aznar-Laín et al., 2023*; *MacKay et al., 2020*]); a newly approved therapy (ganaxolone, a neuroactive steroid that enhances GABAergic inhibition) provides modest improvements but not seizure freedom (*Knight et al., 2022*). Hemizygous males and heterozygous females can be equally and severely affected (*Demarest et al., 2019*; *Wong et al., 2023*). These severe symptoms are in the absence of any structural brain pathology (*Pini et al., 2012*). CDKL5 is a key mediator of synaptic and network development and physiology. Recent analysis of genetic variants in CDD has indicated that CDKL5 kinase function is central to disease pathology (*Demarest et al., 2019*; *Hector et al., 2017b*). In other words, loss of kinase function seems functionally equivalent to loss of the protein.

In rodent models, CDKL5 is expressed throughout the CNS primarily in neurons, increases post-natally, and is stabilized at peak levels in adults (*Chen et al., 2010*; *Hector et al., 2016*; *Baltussen et al., 2018*), indicating roles during development and into adulthood. These features make it impossible to tease apart the precise role of altered CDKL5 function in mediating these symptoms in CDD. It remains unclear whether symptomatic abnormalities are due to chronic CDKL5 dysfunction, acute CDKL5 dysfunction during an earlier critical developmental time point, worsened by epilepsy, or any combination. Adult rodent models of CDD are associated with abnormal behaviors, visual disturbances (*Zhu and Xiong, 2019*), and multiple abnormal signaling cascades (*Wang et al., 2012*; *Fuchs et al., 2015*; *Fuchs et al., 2014*; *Schroeder et al., 2019*; *Ren et al., 2019*). CDKL5 is localized to the dendritic spines of excitatory synapses as well as the nucleus of neurons (*Chen et al., 2010*; *Rusconi et al., 2008*; *Oi et al., 2017*). CDKL5 functions in the formation of excitatory synapses with partners that include PSD-95 (*Zhu et al., 2013*), NGL-1 (*Ricciardi et al., 2012*), Shootin1 (*Nawaz et al., 2016*), and actin (*Chen et al., 2010*). Previous work had suggested CDKL5-specific substrates, including MeCP2, DNMT1, AMPH1, NGL-1, and HDAC4, but reliable antibodies have not been available (*Zhu and Xiong, 2019*). Recent work has identified MAP1S and microtubule end binding protein 2 (EB2) as physiological CDKL5 substrates in brain (*Baltussen et al., 2018*). A phosphospecific antibody for EB2-S222 has been used to report CDKL5 activity in mouse models and human iPSCs (*Baltussen et al., 2018*; *Terzic et al., 2021*; *Di Nardo et al., 2022*). CDKL5 is a binding partner of both PSD-95 and gephyrin (*Uezu et al., 2016*; *De Rosa et al., 2022*); binding with PSD-95 is critical for excitatory spine synapse development (*Wang et al., 2012*) and maintenance of inhibitory synapses (*De Rosa et al., 2022*). Studies have typically found that loss of CDKL5 leads to a global reduction in excitatory synapse numbers (*Ricciardi et al., 2012*; *Della Sala et al., 2016*), reduced PSD-95 (*Negraes et al., 2021*; *Lupori et al., 2019*), and synapsin (*Negraes et al., 2021*) with loss of AMPA-type glutamate receptors (GluA2; *Yennawar et al., 2019*) and increased NMDA-type glutamate receptors (GluN2B; *Okuda et al., 2017*). Inhibitory synapses appear to be unaffected (*Ricciardi et al., 2012*), although inhibitory synaptic currents are affected in some CDKL5 mouse models (*De Rosa et al., 2022*; *Tang et al., 2017*). Embryonic knock-out of *Cdkl5* rodents either enhanced long-term potentiation (LTP) (*Okuda et al., 2017*; *de Oliveira et al., 2022*) or did not affect LTP (*Yennawar et al., 2019*; *de Oliveira et al., 2022*) in an age-dependent fashion. Reports so far do not fully explain the presumed excitation/inhibition imbalance in epilepsy. Further, immature CDKL5-deficient mice do not have seizures or

epilepsy (*Wang et al., 2012*; *Amendola et al., 2014*) and appear insensitive to pro-convulsants such as kainate (*Wang et al., 2012*). This paradox highlights a big gap in our understanding of CDKL5 function: early developmental onset of medically resistant epilepsy in CDD contrasts with a complete lack in experimental models. The reasons for this mismatch are unclear and include possible developmental or other rodent-specific compensations, which typically involve transcriptional changes. Finally, the role of acute CDKL5 kinase dysfunction at any developmental time point is unknown. To address this, we sought to develop a sensitive and specific inhibitor of CDKL5.

*CDKL5* is an X-linked gene encoding a 115 kDa serine-threonine kinase and member of the CMGC family that includes cyclin-dependent kinases (C̲DK), M̲AP-kinases, glycogen synthase kinases (G̲SK), and cyclin-dependent kinase-like (C̲DKL) (*Manning et al., 2002*). It has limited structural homology to CDKL1, 2, 3, and 4 (http://mbv.broadinstitute.org/) but significant homology to GSK3β (*Davis et al., 2011*). *CDKL1-4* mRNA are expressed at much lower levels than *CDKL5* in mouse and human brain (http://mouse.brain-map.org). Of these kinases, only GSK3α (*Shahab et al., 2014*; *Ebrahim Amini et al., 2022*) and GSK3β *Peineau et al., 2008* have been linked to brain synaptic function.

CDKL5 appears to be linked to GSK3β function. *Cdkl5* knock-out mice have increased GSK3β activity, as evidenced by *hypo*-phosphorylation of GSK3β at S-9 and decreased β-catenin (*Fuchs et al., 2015*). β-Catenin, which is phosphorylated and destabilized by active GSK3β, is often used, with S9-phospho-GSK3β as a marker of GSK3β activity (*Wada, 2009*). Inhibition of GSK3β in CDKL5 (young but not old) knock-out mice normalizes expression of β-catenin and S9-phospho-GSK3β (*Fuchs et al., 2015*; *Fuchs et al., 2014*). While upstream signaling is also affected (mTOR and AKT) (*Wang et al., 2012*), it is unclear how CDKL5 and GSK3β activities are linked in CDKL5 knock-out mice. GSK3β mediates a yin-yang interaction of LTP and LTD. Activation of GSK3β is required for the induction of GluN-LTD while inhibition occurs with the induction of LTP (*Peineau et al., 2008*). Thus, development of a specific CDKL5 inhibitor must be careful to exclude cross-talk with GSK3β activity.

Here, we synthesized and characterized specific high-affinity inhibitors of CDKL5 that do not have detectable activity for GSK3β. These compounds are very soluble in water, but blood–brain barrier penetration is low. When applied directly to rat hippocampal brain slices, acute inhibition of CDKL5 selectively reduces postsynaptic function of AMPA-type glutamate receptors in a dose-dependent manner and inhibits a key form of synaptic plasticity, hippocampal LTP. These studies provide new insights into the role of CDKL5 in neuronal function and pathology.

## Results

### Identification of CDKL5 inhibitors that lack GSK3β activity

SNS-032 and AT-7519 were initially identified as potent CDK inhibitors (*Davis et al., 2011*; *Vasta et al., 2018*). We profiled our extensive library of >50 SNS-032 and >100 AT-7519 analogs using the CDKL5 NanoBRET assay. In parallel, the CDKL5 actives ($IC_{50} < 1$ μM) were profiled using the GSK3β NanoBRET assay. NanoBRET is a cellular target engagement assay that relies upon bioluminescence resonance energy transfer (BRET) between a tracer with a red-shifted fluorophore attached and a protein of interest with NanoLuciferase appended. Test compounds, which compete with the tracer for binding to the ATP-binding site, are introduced in a dose-dependent manner and BRET plotted versus concentration, allowing calculation of a target engagement $IC_{50}$ value (*Vasta et al., 2018*). Based on the data generated, a plate of 25 compounds was assembled. The criteria for inclusion on this plate included an $IC_{50} < 400$ nM in the CDKL5 NanoBRET assay and >20fold difference between the CDKL5 NanoBRET and the GSK3β NanoBRET $IC_{50}$ values (*Table 1*).

### A targeted screen of CDKL5 inhibitors in neurons

A set of 20 CDKL5 inhibitors were selected based on their selectivity properties for CDKL5 versus GSK3α/β for testing their inhibition of CDKL5 in rat primary cortical neuron cultures. We tested whether we could detect a dose-dependent reduction of EB2 pSer222 following 1 hr incubation in culture media. We normalized pS222 EB2 with total EB2 levels to assess a specific reduction in phosphorylation as opposed to a loss of total protein levels. Almost all inhibitors showed a pSer222 EB2 reduction at 500 nM; most inhibitors showed a reduction at 50 nM (*Figure 1*). Three inhibitors caused a significant reduction in pSer222 EB2 at 5 nM without a change in total EB2 levels: CAF-382 (**B1**),

**Table 1.** NanoBRET data corresponding to compounds selected for initial study.

| Compound | Parent | CDKL5 NB IC$_{50}$ (nM) | GSK3β NB IC$_{50}$ (nM) | Fold |
|---|---|---|---|---|
| A01 | AT-7519 | 363 | >10,000 | 27.5 |
| A02 | AT-7519 | 178 | 4135 | 23.2 |
| A03 | AT-7519 | 19 | 404.8 | 21.3 |
| A04 | SNS-032 | 108 | 5688 | 52.7 |
| A05 | AT-7519 | 12 | 308.2 | 25.7 |
| A06 | SNS-032 | 221 | 7787 | 35.2 |
| A07 | AT-7519 | 6 | 192 | 32.0 |
| A08 | AT-7519 | 56 | 1374 | 24.5 |
| A09 | AT-7519 | 21 | 481 | 22.9 |
| A10 | SNS-032 | 424 | >10,000 | 23.6 |
| A11 | AT-7519 | 9 | 3699 | 411.0 |
| A12 | AT-7519 | 20 | 524.8 | 26.2 |
| B01 | SNS-032 | 8 | >10,000 | 1250.0 |
| B02 | SNS-032 | 190 | 9100 | 47.9 |
| B03 | SNS-032 | 10 | >10,000 | 1000.0 |
| B04 | AT-7519 | 14 | 553.9 | 39.6 |
| B05 | SNS-032 | 388 | 8000 | 20.6 |
| B06 | AT-7519 | 155 | 8193 | 52.9 |
| B07 | AT-7519 | 36 | 3256 | 90.4 |
| B08 | AT-7519 | 161 | >10,000 | 62.1 |
| B09 | AT-7519 | 37 | 3627 | 98.0 |
| B10 | AT-7519 | 90 | 4945 | 54.9 |
| B11 | AT-7519 | 141 | 3101 | 22.0 |
| B12 | AT-7519 | 69 | 3631 | 52.6 |
| C01 | SNS-032 | 323 | >10,000 | 31.0 |

HW2-013 (**B4**), and LY-213 (**B12**) (*Figure 1*). These represented different chemical parent backbones, so we decided to study these inhibitors further.

## Potency and selectivity analyses of lead compounds

As shown in *Figure 2*, **B1** is an SNS-032 analog, while **B4** and **B12** are analogs of AT-7519. SNS-032 and AT-7519 are published inhibitors of several members of the cyclin-dependent kinase (CDK) and cyclin-dependent kinase-like (CDKL) families (*Davis et al., 2011*). Before using these compounds as tools for dissecting CDKL5 biology, we needed to first understand their kinome-wide selectivity. We analyzed **B1**, **B4**, and **B12** at 1 µM using the Eurofins DiscoverX *scan*MAX panel, which assesses binding to 403 wild-type (WT) human as well as several nonhuman and mutant kinases (*Figure 3*). These cell-free assays employ active site competition to quantitatively measure interactions between a test compound and a kinase. The percent of control (PoC) values are generated for each kinase contained within the panel (*Davis et al., 2011*).

Orthogonal binding or enzymatic assays were employed to confirm the single concentration DiscoverX *scan*MAX panel results above a specified PoC threshold for **B1**, **B4**, and **B12**. A binding assay that employs the Luceome *KinaseSeeker* technology was run for CDKL5 since no commercial enzymatic assay is available. Radiometric enzymatic assays that measure phosphorylation of a validated substrate in the presence of [gamma-33P]-ATP were executed for all other kinases at Eurofins.

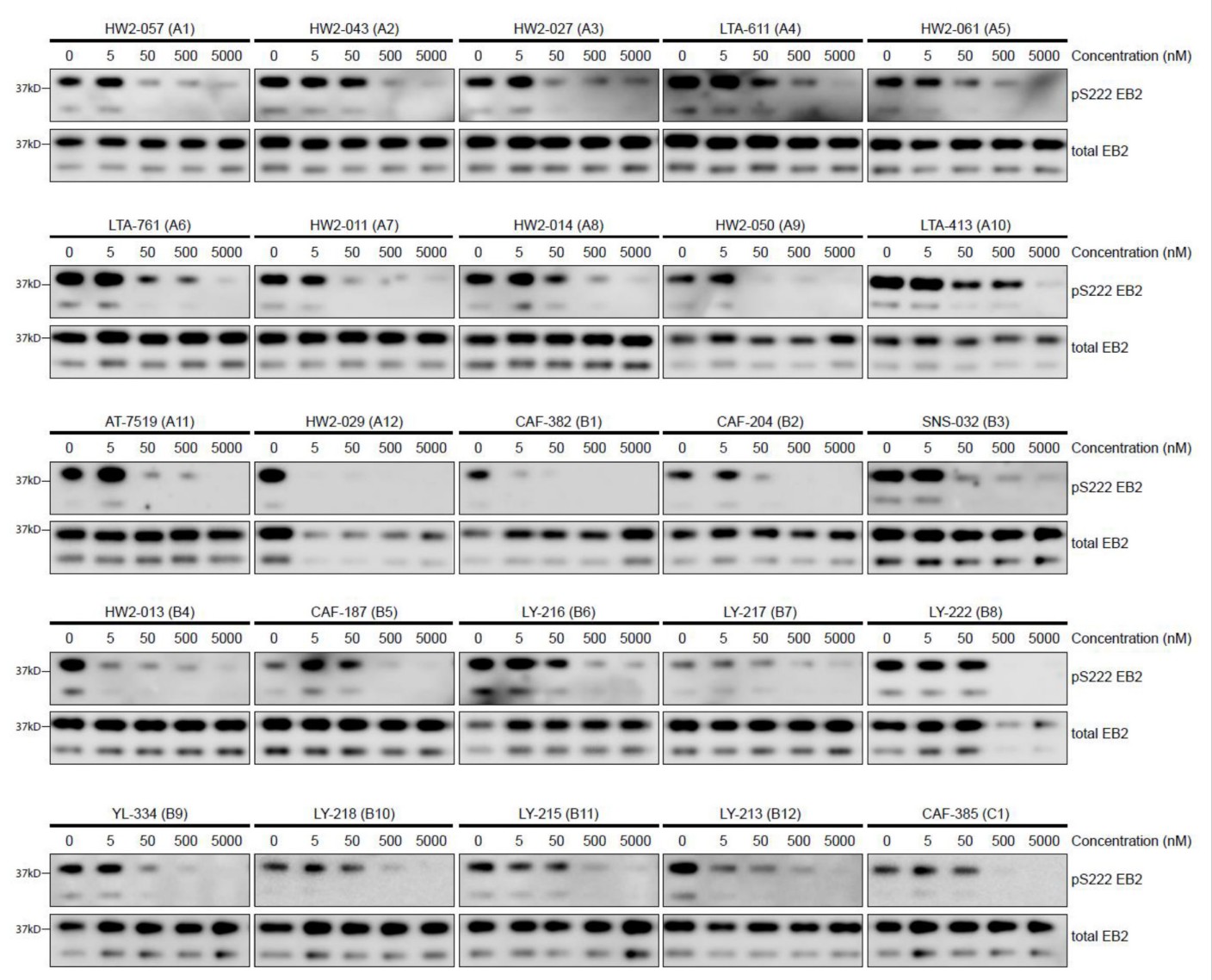

**Figure 1.** Screening of CDKL5 inhibitors in rat primary neurons using western blotting. Western blots showing expression of total EB2 and levels of Ser222 EB2 phosphorylation in DIV14-16 rat primary neurons upon treatment of 1 hr with 5 nM, 50 nM, 500 nM, and 5000 nM of selected CDKL5 inhibitors.

The online version of this article includes the following source data for figure 1:

**Source data 1.** Source data contains raw blots.

Finally, NanoBRET assays corresponding with GSK3α, GSK3β, and CDKL5 were run for all three compounds. Additional NanoBRET profiling was carried out for **B1**.

For **B1**, an analog of SNS-032, pan CDK inhibition was maintained (**Figure 3A**). All kinases with PoC ≤ 20 in the DiscoverX scanMAX panel plus GSK3β were profiled using orthogonal binding or enzymatic assays. While **B1** has a good selectivity score ($S_{10}$ [1 μM] = 0.017; 7 kinases with PoC < 10 at 1 μM), several CDKs (CDK9, PCTK1/CDK16, PCTK2/CDK17, PCTK3/CDK18) are potently inhibited ($IC_{50}$ ≤ 100 nM) (colored green in the nested table in **Figure 3A**). Weaker inhibition of CDK7 as well as both GSK3α and GSK3β was noted. Analysis of biochemical data reveals potent binding/inhibition of CDKL5, CDK9, CDK16, CDK17, and CDK18, and a 44-fold selectivity window between CDKL5 and the next most potently inhibited kinase (CDK7). NanoBRET data reflects a window for potent engagement of CDKL5 at which CDKs are only weakly bound by **B1** (≤100 nM). Curves corresponding to the affinity data for CDKL5, GSK3, and several CDKs are included in **Figure 3—figure supplements 1**

**Figure 2.** Structures of CDKL5 inhibitor leads and corresponding parent compounds.

*and 2A*. As the binding pockets share a high degree of similarly, the selectivity of **B1** within the CDKL family was further profiled via thermal shift and enzymatic assays (*Figure 3—figure supplement 3*). The highest change in melting temperature (ΔTm, >5°C) was noted for CDKL5, while a modest ΔTm of 3–4°C was observed for CDKL2. Radiometric enzyme assays were run for CDKL1–4 to corroborate and expand on this in-family selectivity assessment. When comparing the binding and enzymatic assay results, nearly 100-fold selectivity was observed for CDKL5 versus the most potently inhibited kinase, CDKL2. Inhibition of CDKL3 and CDKL4 by **B1** was modest ($IC_{50}$ = 2.1–2.7 µM), and this compound did not inhibit CDKL1. We propose that only at higher concentrations (>500 nM in a biochemical assay) would inhibition of these kinases become significant, and, based on the data in *Figure 3A*, that an even larger window may exist between CDKL5 and CDKL2 when cells are dosed with **B1**. Importantly, these data support that when cells are dosed with **B1** at ≤100 nM, no CDKL family member other than CDKL5 will be inhibited.

Compounds **B4** and **B12**, which are analogs of AT-7519, also demonstrated good kinome-wide selectivity with $S_{10}$(1 µM) scores of 0.02 and 0.01, respectively (*Figure 3B and C*). These selectivity scores correspond with high-affinity binding (PoC < 10) to eight kinases for **B4** and four kinases for **B12** when profiled at 1 µM. Kinases with PoC < 10 in the DiscoverX *scan*MAX panel when **B4** was screened were profiled using orthogonal binding (CDKL5) or enzymatic (remaining kinases) assays. This orthogonal profiling of **B4** confirmed high-affinity binding to CDKL5, GSK3α, and GSK3β (green in nested table in *Figure 3B*, curves in *Figure 3—figure supplements 1 and 2B*), with slightly weaker affinity for GSK3α/β than CDKL5 evaluated via the respective cellular target engagement (NanoBRET) assays (*Figure 3*, curves in *Figure 3—figure supplements 1 and 2*). Orthogonal profiling of **B12** confirmed high-affinity binding to CDKL5, GSK3α, and GSK3β (green in nested table in *Figure 3C*, curves in *Figure 3—figure supplements 1 and 2C*). This compound is a more effective inhibitor of GSK3α when compared to GSK3β. As was the case for **B4**, **B12** bound with greater affinity to CDKL5 than GSK3α/β in the respective NanoBRET assays (*Figure 3*, curves in *Figure 3—figure supplements 1 and 2C*). Aside from GSK3α/β, a large (>100-fold) window exists between CDKL5 and the next most potently inhibited kinase, DYRK2, when considering the enzymatic and/or binding data.

Binding and orthogonal assay results in combination with western blots confirmed that in cells **B1** is a CDKL5 and multi-CDK inhibitor with much weaker GSK3α/β affinity (>1.8 µM) and inhibitory activity. Compounds **B4** and **B12** exhibited similar inhibition profiles and are potent inhibitors of CDKL5, GSK3α, and GSK3β. Despite its potency, compound **B4** demonstrated suboptimal selectivity when compared with **B1** and **B12**. This was determined based on the number of WT human kinases with PoC < 35 in the DiscoverX *scan*MAX panel when profiled at 1 µM: **B1** = 12, **B4** = 17, and **B12** = 9.

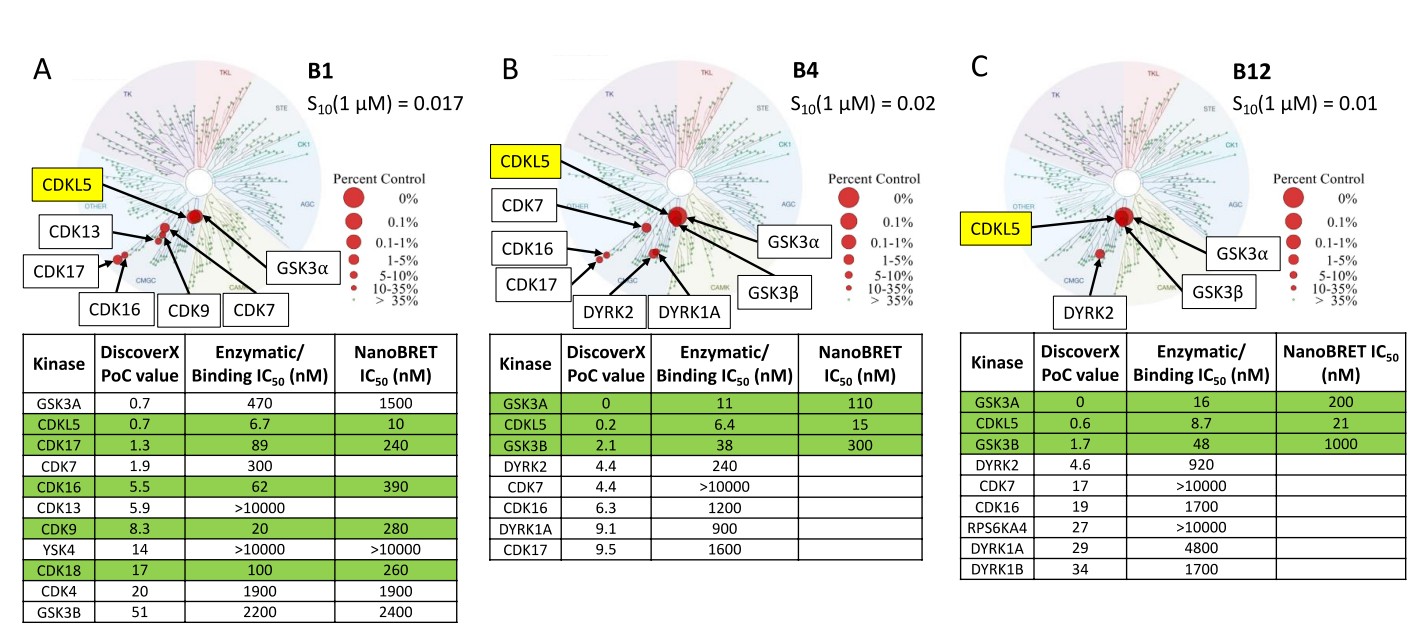

**Figure 3.** Kinome-wide selectivity data for CDKL5 lead compounds. Kinome tree diagrams illustrate the selectivity of these compounds when profiled against 403 wild-type (WT) human kinases at 1 μM at Eurofins DiscoverX in their *scan*MAX panel. Each red circle represents a kinase that binds with high affinity (percent of control [PoC] <10) to the compound being assayed: (**A**) B1, (**B**) B4, and (**C**) B12. These kinases have been labeled for clarity. The percent control legend shows red circles of different sizes corresponding with percent control value each kinase binds the small molecule in this large binding panel. Also included are selectivity scores ($S_{10}$ [1 μM]), which were calculated using the PoC values for WT human kinases in the *scan*MAX panel only. The $S_{10}$ score is a way to express selectivity that corresponds with the percent of the kinases screened that bind with a PoC value <10. In the embedded tables, kinases are listed by their gene names and ranked by their PoC value generated in the *scan*MAX panel. Rows colored green demonstrate enzymatic $IC_{50}$ values within a 30-fold window of the CDKL5 binding $IC_{50}$ value. A binding assay was run only for CDKL5, all other $IC_{50}$ values in the penultimate column of the nested tables were generated using an enzymatic assay (*Figure 3—figure supplements 1 and 2*).

The online version of this article includes the following figure supplement(s) for figure 3:

**Figure supplement 1.** Curves corresponding with CDKL5 affinity measurements in *Figure 3*.

**Figure supplement 2.** Curves corresponding with GSK3 NanoBRET measurements in *Figure 3*.

**Figure supplement 3.** CDKL family selectivity evaluation via thermal shift and binding/enzymatic assays.

## In vitro kinase activity

We employed in vitro kinase assays to evaluate the impact of **B1** on CDKL5 activity (*Figure 4*). After preparing human WT and kinase dead (KD, CDKL5 K42R; *Lin et al., 2005*; *Kim et al., 2020b*) recombinant proteins, equal amounts were employed in CDKL5 activity assays (*Figure 4B*). We observed that, in the presence of ATP (50 μM, *Figure 4A*), WT CDKL5 but not KD CDKL5 elicited kinase activity, as measured by turnover of ATP to ADP via a luminescent reagent (ADP-Glo assay). Next, the activity of untreated WT CDKL5 was compared with WT CDKL5 treated with vehicle (DMSO) or kinase inhibitors (10 or 100 nM). **B1**, AST-487, and lapatinib were the inhibitors chosen. AST-487 is a confirmed, but less selective inhibitor of CDKL5 (positive control) (*Canning et al., 2018*). Lapatinib is a relatively selective kinase inhibitor that does not inhibit CDKL5 (negative control). The difference between DMSO, lapatinib, and untreated CDKL5 was not significant at 10 or 100 nM (*Figure 4C and D*). **B1** and AST-487, however, robustly inhibited CDKL5 in a dose-dependent manner (*Figure 4C and D*). **B1** was confirmed to be a potent inhibitor of CDKL5 kinase activity using this assay.

Kinase domains of GSK3β and CDKL5 have high similarity. To test the cross-talk of these inhibitors with GSK3β, we measured phospho-β-catenin (Ser33/37/Thr41), which is directly phosphorylated by GSK3β (*Baltussen et al., 2018*). We measured pSer222 EB2 levels normalized to total EB2 at increasing concentrations of inhibitors in primary neurons and compared this to phospho-β-catenin normalized to total β-catenin (*Figure 5*). We found that CAF-382 (**B1**) showed a significant reduction in pSer222 EB2/total EB2 at 500 nM concentration (*Figure 5A*). There was a trend of pSer222 EB2/total

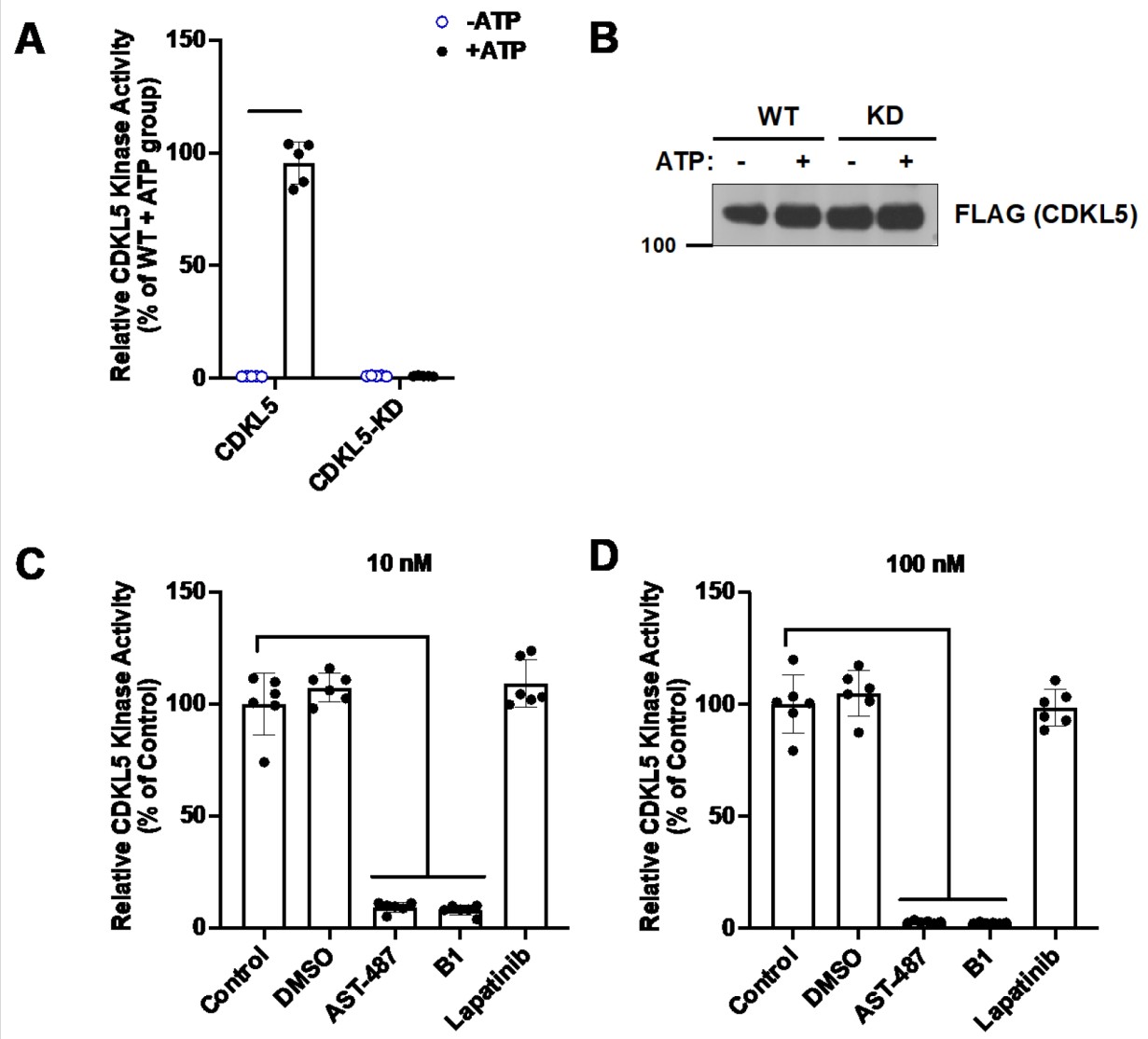

**Figure 4.** B1 potently inhibits the kinase activity of CDKL5. B1 was simultaneously assayed with other CDKL5 antagonists (*Ong et al., 2023*). (**A**) Representative graph from a CDKL5 kinase assay demonstrating that purified WT human CDKL5 retains kinase activity, while the kinase dead (KD, CDKL5 K42R) human protein is functionally inactive (n = 3). (**B**) Representative western blot illustrating that equal expression of WT and KD proteins was observed in the kinase assay experiments. (**C, D**) Kinase assays using purified WT human CDKL5 in the presence of 10 or 100 nM of the indicated compounds (n = 3). One-way ANOVA with Dunnett's multiple-comparison test used. ***p<0.0001 and nonsignificant comparisons not shown. Control, DMSO, positive (AST-487), and negative (Lapatinib) controls are as in *Ong et al., 2023*.

The online version of this article includes the following source data for figure 4:

**Source data 1.** Source data contains raw blots.

EB2 reduction from 5 nM, which was not significant (*Figure 5A*). Phospho-β-catenin/total β-catenin was not changed over the entire concentration range, indicating that CAF-382 (**B1**) does not inhibit GSK3b (*Figure 5A*). HW2-013 (**B4**) was effective in reducing EB2 pSer222 at 500 nM but also inhibited GSK3β significantly at 500 nM with a trend of inhibition from 5 nM, indicating that this inhibitor affects GSK3b (*Figure 5B*). Finally, LY-213 (**B12**) reduced pSer222 EB2 at 500 nM but also seemed to decrease β-catenin phosphorylation at this concentration. LY-213 (**B12**) showed comparable trends of inhibition for GSK3βand CDKL5 (*Figure 5C*). There were no significant differences in total CDKL5, EB2 or β-catenin levels during these treatments (*Figure 5—figure supplement 1*). In support of this assay, phospho-β-catenin (Ser33/37/Thr41) is reduced upon exposure to GSK3β inhibitor CHIR 99021 while

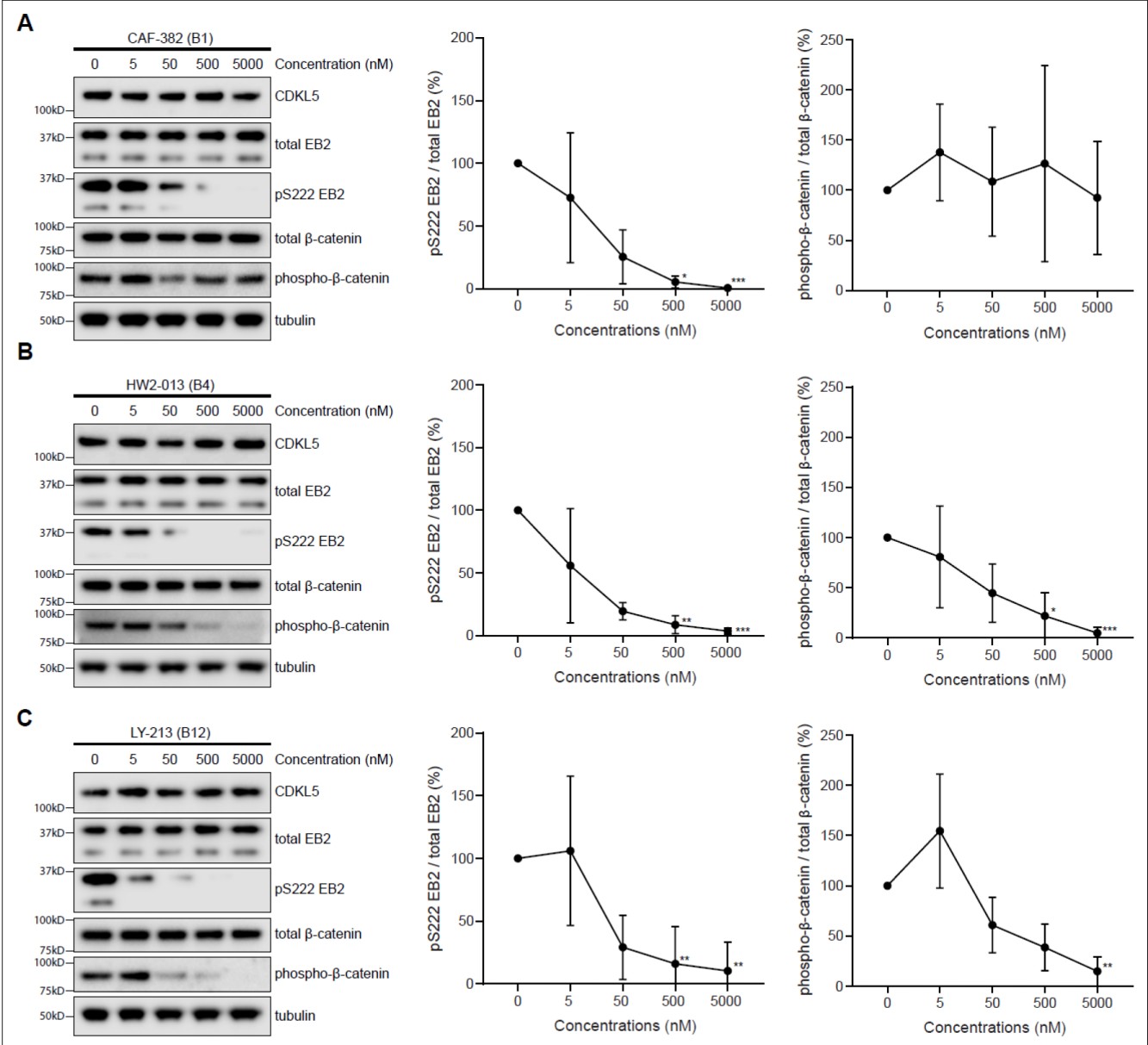

**Figure 5.** CAF-382 (**B1**), HW2-013 (**B4**), and LY-213 (**B12**) compounds reduce CDKL5 activity in rat primary neurons. HW2-013 (**B4**) and LY-213 (**B12**) also downregulate GSK3 activity. (**A**) Western blot and quantification showing expression of EB2 phosphorylation and β-catenin phosphorylation in DIV14-15 rat primary neurons after an hour treatment with different concentrations of CAF-382 (**B1**). (**B**) Western blot and quantification showing expression of EB2 phosphorylation and β-catenin phosphorylation in DIV14-15 rat primary neurons after an hour treatment with different concentrations of HW2-013 (**B4**). (**C**) Western blot and quantification showing expression of EB2 phosphorylation and β-catenin phosphorylation in DIV14-15 rat primary neurons after an hour treatment with different concentrations of LY-213 (**B12**). Each concentration was compared to the control using a Kruskal–Wallis test. n = 3 biological replicates with two repetitions. *p≤0.05; **p≤0.01; ***p≤0.001. Error bars are SD.

The online version of this article includes the following source data and figure supplement(s) for figure 5:

**Source data 1.** Source data contains raw blots.

**Figure supplement 1.** CDKL5, total EB2, and β-catenin expression in rat primary neurons after treatment with CAF-382 (**B1**), HW2-013 (**B4**), and LY-213 (**B12**) compounds.

**Figure supplement 1—source data 1.** Source data contains raw blots.

**Figure supplement 2.** Phosphorylation of β-catenin, a substrate of GSK, is reduced upon treatment with a GSK inhibitor CHIR 99021 (Tocris) but EB2 phosphorylation is not changed.

EB2 pSer222 is not affected by GSK3β inhibition (*Figure 5—figure supplement 2*). We concluded that CAF-382 (**B1**) can inhibit CDKL5 activity without affecting GSK3 activity in neurons.

## Acute CDKL5 inhibition by CAF-382 (B1) in rat hippocampal slices

To determine whether CAF-382 (**B1**) could inhibit CDKL5 activity within intact brain tissue, we prepared acute hippocampal slices from P20–30 rats and incubated them with CAF-382 (**B1**) for 2 hr as described. As for primary cultures, CAF-382 (**B1**) reduced EB2 phosphorylation in a dose-dependent fashion (*Figure 6*). Similarly, SNS-032, the parent compound of CAF-382 (**B1**), also reduced EB2 phosphorylation in hippocampal slices (*Figure 6—figure supplement 1*).

Previous studies found that genetic knock-down of *Cdkl5* in primary rodent brain cultures reduced glutamatergic excitatory synaptic transmission associated with loss of dendritic spines (*Ricciardi et al., 2012*). In this approach, the time course of loss of CDKL5 function is expected to occur over several days. To clarify the more acute role of CDKL5 in regulating excitatory synaptic transmission, we applied CAF-382 (**B1**) (10–100 nM) to hippocampal slices while measuring field excitatory postsynaptic potentials (fEPSPs) in the CA1 hippocampus. While varying the stimulation strength applied to presynaptic Schaffer-collateral pathway inputs, we were able to measure the input–output responsiveness through measuring the input (fiber volley size) versus output (fEPSP slope). The slope of this relationship was reduced in a dose–response relationship by CAF-382 (**B1**) within 30 min of application (*Figure 7A*). In other words, for a similar input (fiber volley amplitude), the postsynaptic responsiveness (fEPSP slope) was reduced (*Figure 7B*) in a time- and dose-dependent fashion (*Figure 7C*). This suggests that the effect of acute CDKL5 inhibition by CAF-382 (**B1**) on glutamatergic synaptic transmission is primarily postsynaptic. Determination that amphiphysin-1, a presynaptic protein, is phosphorylated by CDKL5 (*Sekiguchi et al., 2013*) suggested a role of CDKL5 in presynaptic function. However, *Cdkl5* knock-out studies showed that amphiphysin-1 was not phosphorylated by CDKL5 in vivo, suggesting an alternative amphiphysin-1-independent mechanism is responsible for observed presynaptic changes (*Kontaxi et al., 2023*). To investigate the presynaptic role of CDKL5 kinase function, the paired-pulse ratio across several paired-pulse intervals was measured (*Figure 8*). CAF-382 (**B1**) (100 nM × 1 hr) had no effect on the paired-pulse ratio, confirming the acute effects of CAF-382 (**B1**) are restricted to the postsynapse. fEPSP slopes are mediated by AMPA-type glutamate receptors with NMDA-type glutamate receptors largely silent under these conditions. In order to visualize the impact of CAF-382 (**B1**) on NMDA-type glutamate receptor mediated responses, extracellular $Mg^{2+}$ was removed from the recording medium (*Coan and Collingridge, 1985*). Under these conditions, an NMDA-receptor-mediated field potential (population spike 2) becomes apparent; this was not affected by CAF-382 (**B1**) (*Figure 9*) but was completely blocked by the NMDA-type glutamate receptor antagonist D-APV (50 μM). This result suggests that the acute effect of CAF-382 (**B1**)-mediated CDKL5 inhibition is to selectively reduce AMPA-type glutamate receptor-mediated responses postsynaptically.

A previous study found that germ-line knock-out of *Cdkl5* in the mouse enhanced LTP at 7–13-week-old (wo) hippocampal Schaffer Collateral to CA1 synapses without alteration of the input–output relationship (*Okuda et al., 2017*). In another study, germ-line knock-out of *Cdkl5* in a mouse at 4–5 wo did not affect stable LTP at this synapse (*Yennawar et al., 2019*). In a rat *Cdkl5* knock-out, LTP at this synapse was selectively increased at 3–4 wo but not at later ages (*de Oliveira et al., 2022*). To clarify the more acute role of CDKL5 in regulating LTP at this synapse, we incubated 3–4 wo hippocampal slices with CAF-382 (**B1**) (100 nM) for at least 30 min and, following stable baseline measurements of fEPSP slope, induced LTP via theta burst to Schaffer Collateral to CA1 synapses in the continuous presence of CAF-382 (**B1**). Compared to interleaved recordings from control slices, CAF-382 (**B1**) significantly reduced LTP at 60 min post theta-burst (*Figure 10*). To investigate this further, LTP at these synapses of isolated AMPA-type glutamate receptors via a pairing protocol was measured using whole-cell patch-clamp. In this case, hippocampal slices were incubated with CAF-382 (**B1**) (100 nM) for only 1 hr prior to, but not during, recording. Compared to interleaved recordings from control slices, CAF-382 (**B1**) significantly reduced LTP at 30 min post pairing (*Figure 11*). The induction and expression of LTP include the contributions of presynaptic and postsynaptic mechanisms at both excitatory and inhibitory synapses (*Collingridge and Abraham, 2022*; *Nicoll, 2017*). Since we do not see a presynaptic effect of CAF-382 (**B1**) (*Figure 8*), and the approach (*Figure 11*) removed the contribution of inhibitory synapses, this supports the direct role of CDKL5 in the postsynaptic mediation of LTP at this synapse at this age.

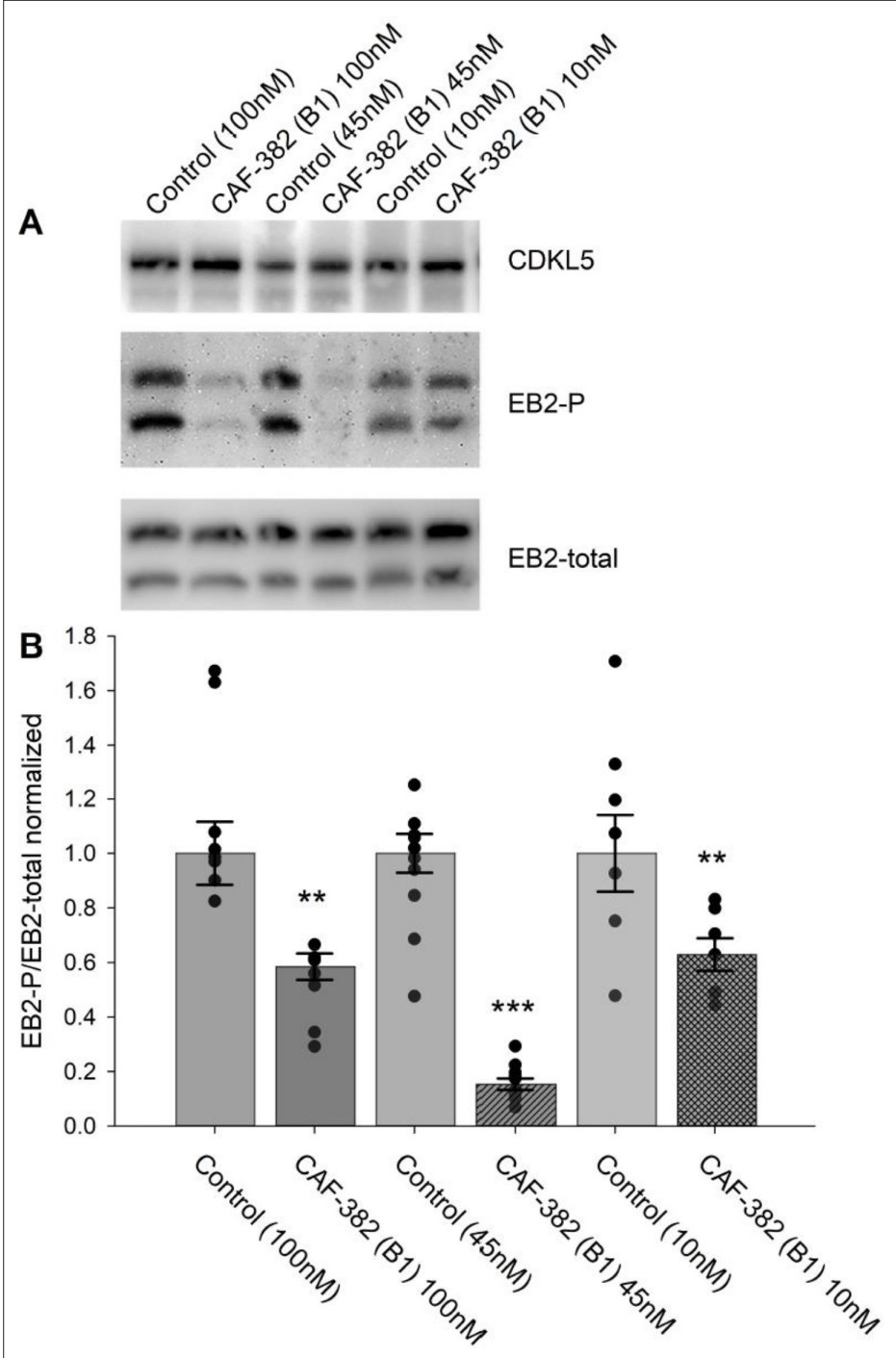

**Figure 6.** CAF-382 (**B1**) reduced phosphorylation of EB2 in CA1 hippocampal slices. (**A**) Example blots demonstrate no alterations in CDKL5 expression across treatments (stats in text) and suggest relative CAF-382 (**B1**) dose-dependent differences in EB2 phosphorylation. (**B**) Normalized quantification of EB2-phosphorylation (density of EB2-phosphorylation bands/ density of EB2-total bands). CAF 382 (**B1**) (control: 1 ± 0.10 vs. 100 nM: 0.49 ± 0.02, n = 8, p=0.002, RM-ANOVA) (control: 1 ± 0.08 vs. 45 nM: 0.16 ± 0.03, n = 10, p<0.001, RM-ANOVA) (control: 1 ± 0.14 vs. 10 nM: 0.62 ± 0.11, n = 7, p=0.003, RM-ANOVA) reduced EB2 phosphorylation at all concentrations. *p≤0.05; **p≤0.01; ***p≤0.001. Error bars are SE.

The online version of this article includes the following source data and figure supplement(s) for figure 6:

**Source data 1.** Source data contains raw blots.

*Figure 6 continued on next page*

*Figure 6 continued*

**Source data 2.** Source data contains raw blots.

**Source data 3.** Source data contains raw blots.

**Figure supplement 1.** SNS-032 (Selleck Chemicals) inhibits phosphorylation of EB2 in rat hippocampal slices in a time- and concentration-dependent fashion.

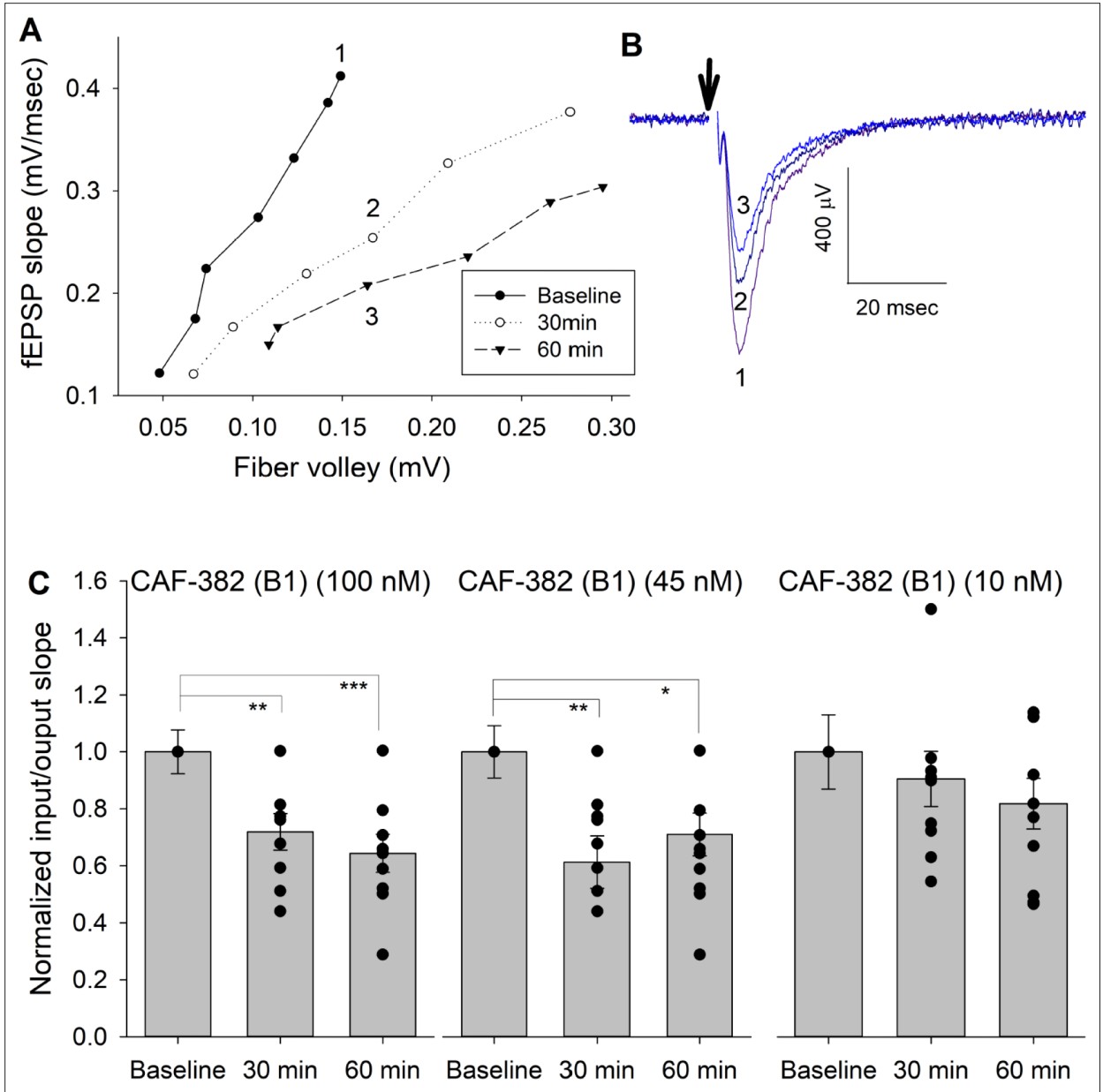

**Figure 7.** CAF-382 (**B1**) reduced postsynaptic fEPSP responsiveness, as measured by input–output curves. (**A**) Sample input (fiber volley)/output (fEPSP slope) (I/O) curves at baseline, 30 min and 60 min after CAF-382 (**B1**) (100 nM) in a hippocampal slice. Slopes of I/O curves were obtained at baseline, and 30 min and 60 min after treatment with CAF-382 (**B1**). Numbers in (**A**) indicate sample traces in (**B**) from treated slice; stimulus artifact (arrow) has been removed for clarity. Initial negative deflection after the artifact is the fiber volley, followed by the fEPSP. (**C**) RM-ANOVA of I/O slopes of CAF-382 (**B1**) (100 nM and 45 nM) were significantly decreased after 30 min (100 nM: 1.91 ± 0.23, n = 9, p=0.006; 45 nM: 1.58 ± 0.24, n = 10, p=0.013) and 60 min (100 nM: 2.00 ± 0.30, n = 9, p=0.001; 45 nM: 1.57 ± 0.22, n = 10, p=0.009) compared to baseline (100 nM: 3.24 ± 0.52, n = 9; 45 nM: 2.29 ± 0.23, n = 10). Baseline I/O slope was used to normalize each recording to allow comparisons across all treatments. CAF-382 (**B1**) (10 nM) did not alter I/O slopes after 30 or 60 min (RM-ANOVA, n = 9, p=0.07). *p≤0.05; **p≤0.01; ***p≤0.001. Error bars are SE. fEPSP, field excitatory postsynaptic potentials.

**Table 2.** Solubility and microsomal stability data for **B1**.

| Compound | Kinetic solubility | | Mouse liver microsomal stability (%) | | |
|---|---|---|---|---|---|
| | µM | µg/mL | T = 0 min | T = 30 min | T = 30 min (-NADPH) |
| B1 | 196.8* | 72.1* | 100 | 86.1 | 87.5 |

*Measured solubility is >75% dose concentration, actual solubility may be higher.

## Evaluation of suitability of CDKL5 lead for in vivo studies

As our studies were all executed in vitro, the suitability of **B1** for use in vivo was unknown. To gauge whether **B1** could be dosed in animals using a water-based formulation, we first analyzed its aqueous kinetic solubility (*Table 2*). **B1** demonstrated good solubility (196 µM), supporting that it readily dissolves in aqueous buffer without observed precipitation. Mouse liver microsomal stability data was also collected (*Table 2*). **B1** proved to be very tolerant to microsomal exposure, demonstrated excellent stability (>85%) after 30 min, and was not prone to non-NADPH-dependent enzymatic degradation. This stability is not true for other kinase inhibitor scaffolds, which are significantly degraded after just 30 min (*Yang et al., 2023*). Since compound **B1** was found to have the required aqueous solubility and mouse liver microsomal stability, it was submitted for in vivo characterization. Pharmacokinetic analysis and brain exposure of mice following intraperitoneal administration with **B1** were next examined. Plasma was sampled 0.5, 1, 2, and 4 hr post-dose with 2.29 mg/kg of a salt form of **B1** (*Figure 12*). No abnormal clinical symptoms were observed in treated animals. The half-life of **B1** was found to be <1 hr, with the highest concentration of drug in the plasma found 0.5 hr after dosing (Cmax = 508 ng/mL) and total system exposure (AUCinf) of 646 h*ng/mL (*Figure 12*). The same dose (2.29 mg/kg) as well as a 7.63 mg/kg dose were used to analyze the brain penetration of **B1** in mice 1 hr after administration. As shown in *Table 3*, brain and plasma sampling indicated that brain penetration of **B1** was generally low.

## Discussion

CDD is a rare, devastating, genetically mediated neurological disorder with symptomatic onset in very early infancy. Though there are newly approved therapies that provide some relief for epilepsy (*Knight et al., 2022*), the additional features that are impactful to caregivers (*Neul et al., 2023*) of motor, cognitive, visual, and autonomic disturbances [*Bahi-Buisson et al., 2012*; *Bahi-Buisson et al., 2008*; *Nemos et al., 2009*; *Castrén et al., 2011*; *Melani et al., 2011*; *Fehr et al., 2015*] are not currently addressed. CDKL5 protein (*Trazzi et al., 2018*) and *CDKL5* gene (*Gao et al., 2020*) replacement approaches are advancing through preclinical studies to potential first-in-human trials (https://www.ultragenyx.com/our-research/pipeline/ux055-for-cdd/). While CDKL5 is a key mediator of synaptic and network development and physiology, the precise role of this kinase as explored through genetic manipulations in rodent models remains unclear. This knowledge gap prevents a full understanding of the role of CDKL5 across development and has implications for determining and interpreting when protein and gene replacement strategies may be most efficacious. As CDKL5 kinase function is central to disease pathology (*Demarest et al., 2019*; *Hector et al., 2017b*), our development of a specific CDKL5 kinase inhibitor aims to address this gap.

Following identification of existing inhibitors (*Vasta et al., 2018*), detailed analysis of the activity of analogs of these inhibitors in a range of assays identified three compounds with significant selectivity

**Table 3.** Brain exposure results for **B1**.
(**A**) Brain/plasma concentration–time data after 2.29 mg/kg dose. (**B**) Brain/plasma concentration–time data after 7.63 mg/kg dose.

| A | IP dose: 2.29 mg/kg | | | B | IP dose: 7.63 mg/kg | | |
|---|---|---|---|---|---|---|---|
| Time (hr) | Brain concentration (ng/g) | Plasma (ng/mL) | Ratio (brain/plasma) | Time (hr) | Brain conc entration (ng/g) | Plasma (ng/mL) | Ratio (brain/plasma) |
| 1 | 15.6 | 340 | 0.0459 | 1 | 158 | 750 | 0.210* |

*One animal artificially drove up blood/plasma ratio. N = 3 per dose.

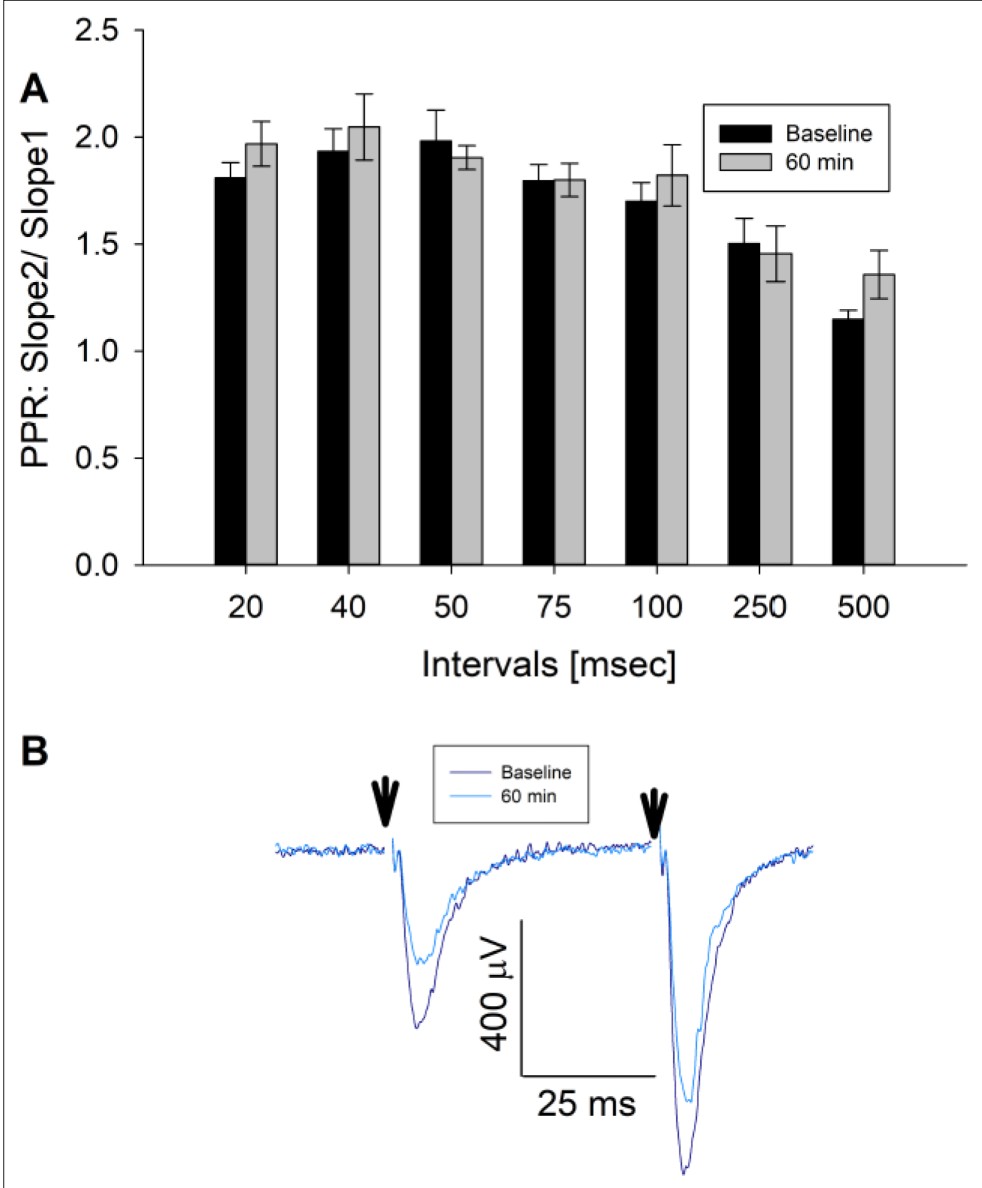

**Figure 8.** CAF-382 (B1) did not alter presynaptic release as measured by paired-pulse ratio (PPR). (**A**) PPRs of fEPSPs (slope of fEPSP2/Slope of fEPSP1), reflective of presynaptic release, were unaltered across a range of stimulus intervals (RM-ANOVA, n = 7) by CAF-382 (**B1**) (100 nM) for 60 min compared to baseline. (**B**) Sample trace for 50 ms interval. Reduction of initial fEPSP slopes were consistent with *Figure 7*. Stimulus artifacts (arrows) have been removed for clarity. Error bars are SE. fEPSP, field excitatory postsynaptic potentials.

over GSK3β, a key requirement due to structural homology (*Davis et al., 2011*) and linkage of GSK3β to both synaptic physiology (*Peineau et al., 2008*) and CDKL5-related signaling (*Fuchs et al., 2015*; *Fuchs et al., 2014*). Like their parent inhibitors, SNS-032 for **B1** and AT-7519 for **B4** and **B12**, the selectivity profiling of these analogs revealed them to retain potent affinity for several CMGC family kinases. The lead compound, **CAF-382 (B1)**, demonstrated potent activity in multiple assays, including target-based in vitro biochemical assays and functional assays utilizing both rodent neuronal cultures and brain slices. The results across all assays are consistent and align with the hypothesis that compound **B1** inhibits CDKL5 and blocks phosphorylation of a known CDKL5 substrate, EB2 (*Baltussen et al., 2018*; *Terzic et al., 2021*; *Di Nardo et al., 2022*). Several CDKs that are potently (IC$_{50}$ ≤ 100 nM) inhibited (CDK9 [*Yu and Cortez, 2011*], PCTK1/CDK16 [*Mikolcevic et al., 2012*], PCTK2/CDK17 [*Liu et al., 2017*], and PCTK3/CDK18 [*de Oliveira Pepino et al., 2021*]) by **B1** are expressed in the brain

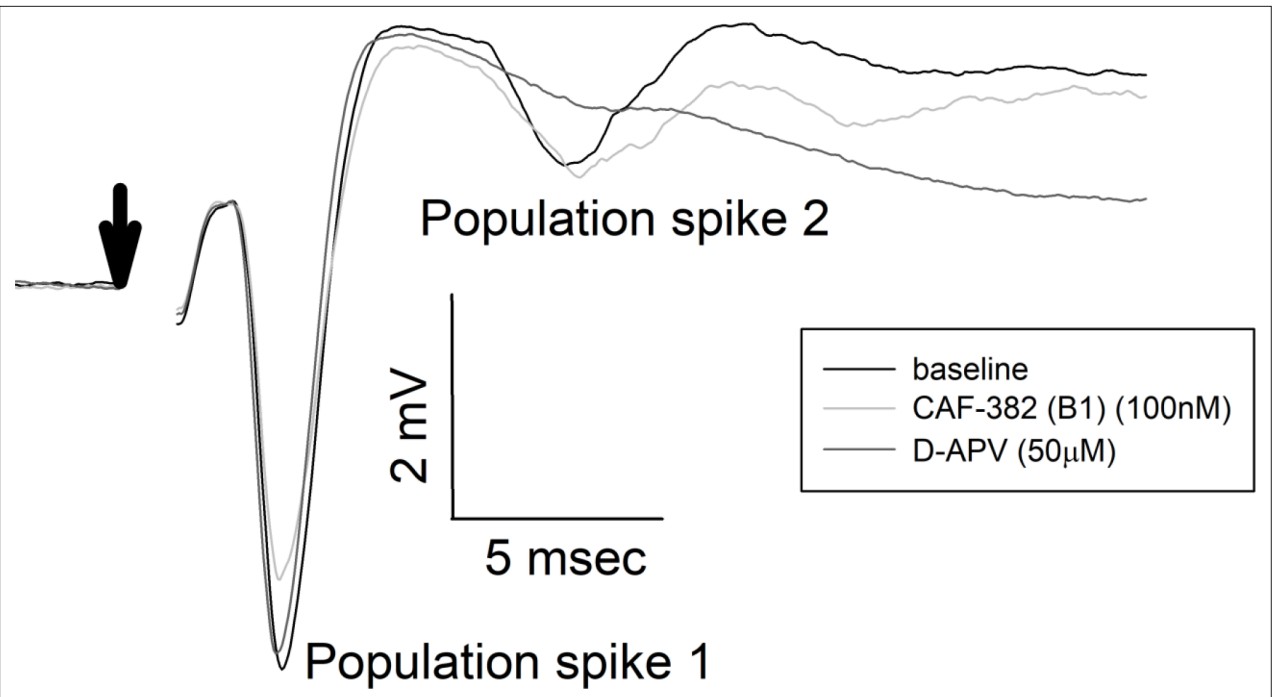

**Figure 9.** CAF-382 (**B1**) did not alter NMDAR-mediated component of fEPSPs. Maximal fEPSPs recorded in stratum pyramidale of CA1 with reduced extracellular $Mg^{2+}$ demonstrate multiple population spikes; previous studies have demonstrated that secondary population spikes in these conditions are NMDAR-dependent (**Coan and Collingridge, 1985**). CAF-382 (**B1**) (100 nM) did not alter population spike 1 (control –4.057 ± 0.584 mV, CAF-382 (**B1**) 3.929 ± 0.736 mV, n = 10, p=0.742, RM-ANOVA) or NMDAR-mediated population spike 2 (control –2.093 ± 0.568 mV, CAF-382 (**B1**) –1.656 ± 0.4442, n = 10, p=0.867, RM-ANOVA). For comparison, the selective NMDAR antagonist D-APV (50 µM) completely blocked population spike 2, as previously reported (**Coan and Collingridge, 1985**). Stimulation artifact has been removed (arrow) for clarity. fEPSP, field excitatory postsynaptic potentials.

(http://mouse.brain-map.org). However, none of these have been demonstrated to have synaptic activity or localization; each are largely involved in replication and nuclear functions. Furthermore, the NanoBRET data for **B1** (**Figure 3**) supports that it is highly specific for CDKL5 when used at a proper concentration in a cellular context. Based on the NanoBRET $IC_{50}$ values for CDKL5 (10 nM) versus off-target CDKs (240–390 nM), at concentrations of ≤100 nM only CDKL5 is engaged by **B1** in cells. Thus, the effects of **B1** as used here are likely highly selective for CDKL5.

The biochemical activity of **CAF-382** (**B1**) against CDKL5 is <5 nM, based on in vitro activity studies (**Figure 4**), and for studies investigating cellular target engagement as well as activity in neuronal cultures and brain slices at the specific substrate, EB2, and synaptic physiology we observe a tenfold shift in potency (~50 nM). A loss in potency, albeit modest, when moving from a biochemical to cell-based assay is commonly observed, mostly due to suboptimal cell penetrance of the small molecule, and has been observed for several orthogonal chemical series of kinase inhibitors (**Drewry et al., 2022a**; **Wells et al., 2021**; **Drewry et al., 2022b**). As the in vitro assay eliminates the contribution of phosphatases (and the phosphatases specific for EB2 are currently unknown), the apparent mismatch could also be due to residual presence of phosphatase-dependent and previously phosphorylated EB2 balanced with active phosphorylation by CDKL5. Nevertheless, CDKL5 kinase function is clearly necessary for the induction and/or expression of LTP, adding to the very short list of kinases required for this essential process linked to learning and memory (**Collingridge and Abraham, 2022**; **Nicoll, 2017**; **Giese and Mizuno, 2013**) and finally clarifying previous genetic studies. The increased LTP seen in genetic studies suggests that observations of increased LTP in these models are compensatory and not directly due to loss of CDKL5. One interpretation of those prior studies is that CDKL5 provided a limiting 'brake' on LTP, with genetic loss resulting in more LTP, which is in contradistinction to our findings. The role of CDKL5 at inhibitory synapses was not specifically examined. Nevertheless, synaptic function at the developmental age studied (P20–30) requires CDKL5 activity for synaptic plasticity, suggesting that strategies that boost function at this age-equivalent in humans may be clinically important. Selective stability and plasticity-dependent insertion of AMPA-type glutamate receptors at

CA1 hippocampal synapses likely require a balance between active CDKL5 kinase and phosphatases. Further studies are needed to determine this balance more specifically and continue to clarify CDKL5 function at presynapses (*Kontaxi et al., 2023*).

While CAF-382 (**B1**) was well tolerated by mice, its short half-life (<1 hr, *Table 1*) would make frequent dosing a necessity. For in vitro studies, however, this compound has demonstrated excellent selectivity and potency in cells as well as brain slices. CAF-382 (**B1**) is one of the best available chemical probes for interrogating CDKL5 activity in vitro, where we envision the greatest utility for future studies. While no clinical abnormalities were observed with in vivo administration of CAF-382 (**B1**), this is likely due to low brain penetration. Due to low brain penetration of CAF-382 (**B1**), strategies that optimize brain penetration or different administration routes (such as direct brain infusion via an implantable osmotic pump) could be considered in future studies investigating the role of CDKL5 in vivo. A hit-to-lead campaign would be one approach to improve the pharmacokinetic and brain penetrance of improved analogs of CAF-382 (**B1**), furnishing others that may be better suited to study the role of CDKL5 in CDD in vivo. These studies, behavioral and perhaps with electrical assessments of seizure activity, may clarify the apparent mismatch of rodent knock-out with clinical observations. Using CDKL5 inhibitors both in vitro as done here and in vivo across developmental stages can help ascertain possible therapeutic windows for CDD. More broadly, CDKL5 inhibition is beneficial for cell survival upon ischemia or nephrotoxin-induced kidney injury (*Kim et al., 2020b*; *Kim et al., 2020a*), suggesting that the inhibitors we characterized here may prove useful to evaluate the promise of CDKL5 inhibition in other disease models.

# Materials and methods

## Key resources table

| Reagent type (species) or resource | Designation | Source or reference | Identifiers | Additional information |
|---|---|---|---|---|
| Cell line (human) | HEK293 | ATCC | CRL-1573 | Routinely subjected to extensive quality control and physiological and molecular characterization (STR profiling). Mycoplasma negativity is regularly confirmed. |
| Antibody | Anti-CDKL5 (sheep polyclonal) | University of Dundee | Anti-CDKL5 350-650 | 1:2000 |
| Antibody | Anti-pS222 EB2 (rabbit polyclonal) | Covalab, from *Baltussen et al., 2018* | | 1:1000-1:2000 |
| Antibody | Anti-EB2 (rat polyclonal) | Abcam | ab45767 | 1:1000 |
| Antibody | Anti-FLAG (rabbit polyclonal) | Cell Signaling | 14793 | 1:1000 |
| Antibody | Anti-rabbit IgG HRP (goat polyclonal) | Jackson ImmunoResearch | 111-035-144 | 1:5000 |
| Antibody | Anti-tubulin (mouse monoclonal) | Sigma | T9026 | 1:100,000 |
| Antibody | Anti-CDKL5 (rabbit polyclonal) | Atlas | HPA002847 | 1:1000 |
| Antibody | Anti-β-catenin (rabbit polyclonal) | Cell Signaling | 9562 | 1:1000 |
| Antibody | Anti-phospho-β-catenin (Ser33/37 Thr41) (rabbit polyclonal) | Cell Signaling | 9561 | 1:250 |
| Commercial assay or kit | CDKL5 TE Assay | Promega | NLuc-CDKL5 (NV2911); Transfection Carrier DNA; NanoBRET TE Intracellular Kinase Assay, K-11 (N2650) | |
| Commercial assay or kit | GSK3α TE Assay | Promega | NLuc-GSK3A (NV3191); Transfection Carrier DNA; NanoBRET TE Intracellular Kinase Assay, K-8 (N2620) | |

*Continued on next page*

*Continued*

| Reagent type (species) or resource | Designation | Source or reference | Identifiers | Additional information |
|---|---|---|---|---|
| Commercial assay or kit | GSK3β TE Assay | Promega | NLuc-GSK3B (NV3201); Transfection Carrier DNA; NanoBRET TE Intracellular Kinase Assay, K-8 (N2620) | |
| Commercial assay or kit | CDK9/Cyclin K TE Assay | Promega | NLuc-CDK9 (NV2871); CCNK Expression Vector (NV2661); NanoBRET TE Intracellular Kinase Assay, K-8 (N2620) | |
| Commercial assay or kit | CDK16/Cyclin Y TE Assay | Promega | NLuc-CDKL16 (NV2741); CCNY Expression Vector (NV2691); NanoBRET TE Intracellular Kinase Assay, K-12 (NF1001) | |
| Commercial assay or kit | CDK17/Cyclin Y TE Assay | Promega | NLuc-CDKL17 (NV2751); CCNY Expression Vector (NV2691); NanoBRET TE Intracellular Kinase Assay, K-12 (NF1001) | |
| Commercial assay or kit | CDK18/Cyclin Y TE Assay | Promega | NLuc-CDKL18 (NV2761); CCNY Expression Vector (NV2691); NanoBRET TE Intracellular Kinase Assay, K-12 (NF1001) | |
| Commercial assay or kit | *scan*MAX | Eurofins DiscoverX Corporation | | |
| Commercial assay or kit | *KinaseSeeker* | Luceome Biotechnologies | | CDKL5 only |
| Commercial assay or kit | Radiometric kinase assays | Eurofins DiscoverX Corporation | | |
| Chemical compound, drug | **B1** (CAF-382) | This paper, *Figures 1–5 and 7* | | Available from senior authors (alison.axtman@unc.edu) |
| Chemical compound, drug | **B4** (HW2-013) | This paper, *Figures 1–3, Figures 7 and 14* | | Available from senior authors (alison.axtman@unc.edu) |
| Chemical compound, drug | **B12** (LY-213) | This paper, *Figures 1–3, Figures 7 and 14* | | Available from senior authors (alison.axtman@unc.edu) |

## Synthesis of key compounds and chemistry general information

Reagents were purchased from reputable commercial vendors, including Sigma-Aldrich, Combi Blocks, Enamine, and others, and used in accordance with safety data sheets. A rotary evaporator was employed to remove solvent(s) under reduced pressure (in vacuo). Thin-layer chromatography (TLC) as well as LC–MS were used to monitor reaction progress. The following abbreviations are used in experimental procedures: mmol (millimoles), µmol (micromoles), µL (microliters), mg (milligrams), equiv (equivalent(s)), min (minutes), and hr (hours). Compound **B1** was prepared as a TFA salt with >95% purity (by HPLC) according to Scheme 1 (*Figure 13*) via a reductive coupling reaction in the presence of sodium borohydride and ethanol followed by amide coupling and deprotection. Compounds **B4** and **B12** were prepared as an HCl salt with >95% purity (by NMR) according to Scheme 2 (*Figure 14*). An amide coupling reaction in the presence of propylphosphonic anhydride (T3P) followed by nitro group reduction generated a common intermediate. Subsequent amide coupling reaction followed by deprotection yielded **B4** and **B12**. Full characterization of the intermediates and final products (*Figure 2*) is included as (*Figure 13—figure supplement 1* and *Figure 14—figure supplements 1 and 2*). $^1$H NMR and $^{13}$C NMR spectra as well as microanalytical data were collected for a key intermediate and final compound **B1** to confirm their identity and evaluate their purity prior to initiating cell- and animal-based studies. $^1$H and $^{13}$C NMR spectra were collected in DMSO-$d_6$ using Bruker spectrometers. Magnet strength is indicated in each corresponding line listing. Peak positions are included in parts per million (ppm) and calibrated versus the shift of DMSO-$d_6$; coupling constants (*J* values) are noted in hertz (Hz); and multiplicities are included as follows: singlet (s), doublet (d), pentet of doublets (pd), and multiplet (m).

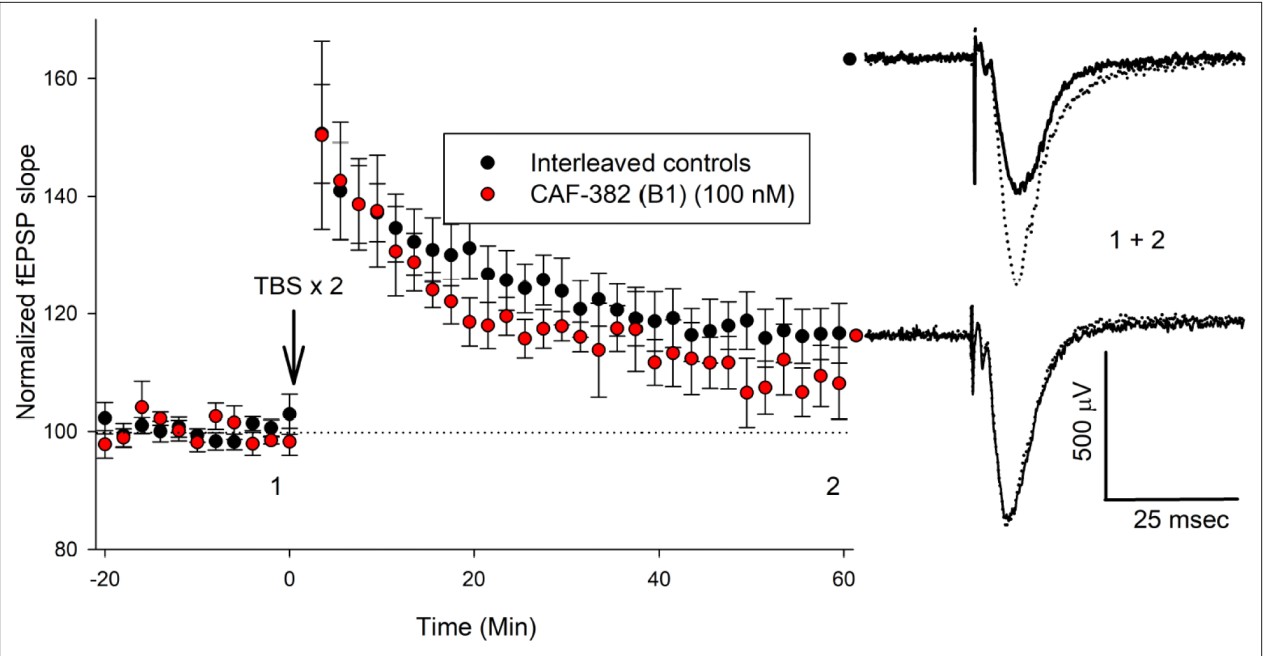

**Figure 10.** CAF-382 (**B1**) acutely reduces the expression of long-term potentiation (LTP) in CA1 hippocampus. Hippocampal slices were initially incubated in CAF-382 (**B1**) for at least 30 min and then continuously perfused with CAF-382 (**B1**) (100 nM). fEPSP slope was normalized to initial baseline, expressed as 100% (dotted line); mean ± SEM shown. Following stable baseline of fEPSP slope (>20 min), LTP was induced by two theta-burst (TBS) trains. Compared to interleaved control slices, CAF-382 (**B1**) significantly reduced LTP as measured by fEPSP slope at 56–60 min post theta-burst (control: 115.6 ± 2.2 n = 13 slices, 12 rats; CAF-382 (**B1**): 106.4 ± 3.5, n = 5 slices, four rats; p=0.029, two-way ANOVA, Holm–Sidak post hoc). Sample traces for control (black filled circle) and CAF-382 (**B1**) (red filled circle) before (1, solid trace) and after LTP (2, dotted trace) are shown to the right. Error bars are SE. fEPSP, field excitatory postsynaptic potentials.

## NanoBRET assays

Human embryonic kidney (HEK293) cells obtained from ATCC (Manassas, VA) were cultured in Dulbecco's Modified Eagle's medium (DMEM, Gibco) supplemented with 10% (v/v) fetal bovine serum (FBS, Corning). These cells were incubated at 37°C in 5% $CO_2$ and passaged every 72 hr with trypsin (Gibco) so that they never reached confluency. Promega (Madison, WI) kindly provided full-length, human constructs for NanoBRET measurements of CDKL5 (NLuc-CDKL5; hCDKL5_1), GSK3α (NLuc-GSK3α), GSK3β (NLuc-GSK3β), CDK9 (NLuc-CDK9), CDK16 (NLuc-CDK16), CDK17 (NLuc-CDK17), and CDK18 (NLuc-CDK18) included in *Table 1* as well as in *Figure 3*, *Figure 3—figure supplement 1*, and *Figure 3—figure supplement 2*. The fusion vector sequences for each can be found on the Promega website. *hCDKL5 and rCdkl5* isoforms are homologous through the kinase domain (*Hector et al., 2017b*; *Hector et al., 2016*; *Hector et al., 2017a*). N-terminal NLuc orientations were used for all seven kinases. NanoBRET assays were executed in dose–response (12-pt curves) format in HEK293 cells as previously reported (*Wells et al., 2021*; *Drewry et al., 2022b*; *Wells et al., 2020*). Assays were executed as recommended by Promega, using 0.31 µM of tracer K11 for CDKL5, 0.13 µM of tracer K8 for GSK3α, 0.063 µM of tracer K8 for GSK3β, 0.063 µM of tracer K8 for CDK9, 0.25 µM of tracer K12 for CDK16, 0.13 µM of tracer K12 for CDK17, and 0.13 µM of tracer K12 for CDK18. Cyclin K was transfected into the cells for the CDK9 NanoBRET assay, while cyclin Y was transfected into the cells for the CDK16, CDK17, and CDK18 NanoBRET assays. Where shown, error bars represent SD.

## Analysis of kinome-wide selectivity

The broad selectivity of compounds **B1**, **B4**, and **B12** at 1 µM was evaluated via the Eurofins DiscoverX Corporation (Fremont, CA) *scan*MAX assay platform, the largest commercial kinase panel available. The *scan*MAX assays are cell-free, ATP-free, and employ proprietary active site competition to quantitatively measure binding between a test compound and a kinase. Different kinase constructs, truncates or full-length, are used to enable different assays for mutant, lipid, and atypical human kinases as well as pathogen kinases. Since these assays do not require ATP, values generated are reflective of

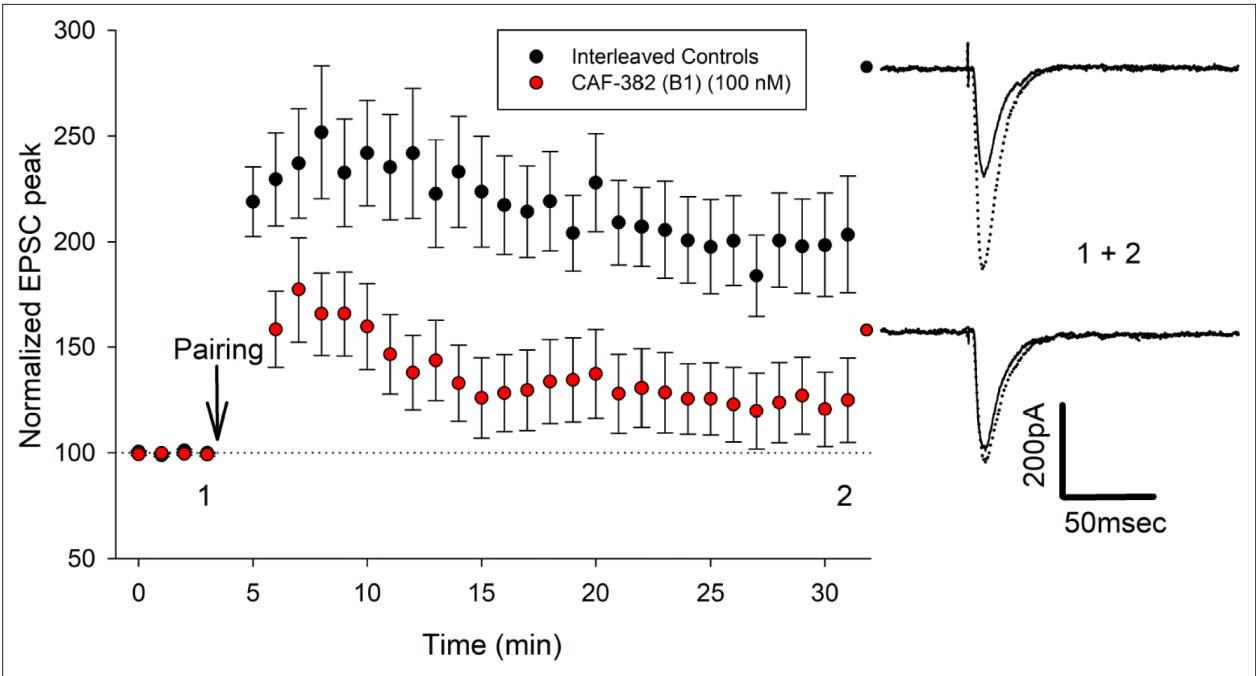

**Figure 11.** CAF-382 (**B1**) acutely reduces the expression of long-term potentiation (LTP) mediated by AMPA-type glutamate receptors in CA1 hippocampus. Hippocampal slices were initially incubated in CAF-382 (**B1**) (100 nM) for at least 60 min prior to recording. Peak negative current of AMPA-type glutamate receptor-mediated synaptic responses were normalized to initial baseline post break-in (time = 0), expressed as 100% (dotted line); mean ± SEM shown. Following baseline, LTP was induced by a pairing protocol. Compared to interleaved control slices, CAF-382 (**B1**) significantly reduced LTP at 27–31 min post break-in (control: 197.5 ± 8.4 n = 14 slices, 14 rats; CAF-382 (**B1**): 123.14 ± 9.0, n = 12–13 slices, 13 rats; p<0.001, two-way ANOVA, Holm–Sidak post hoc). Sample traces for control (black filled circle) and CAF-382 (**B1**, red filled circle) before (1, solid trace) and after LTP (2, dotted trace) are shown to the right. Error bars are SE.

thermodynamic interaction affinities and span a large dynamic affinity range. In total, 403 wild-type (WT) human kinases were included in these analyses. Percent of control (PoC) values were generated, which allowed for the calculation of selectivity scores ($S_{10}$ [1 µM]) noted in *Figure 3* (*Davis et al., 2011*). Selectivity scores ($S_{10}$ [1 µM]) were calculated using the PoC values for only WT human kinases in the DiscoverX *scan*MAX panel. The $S_{10}$ score expresses selectivity and corresponds with the percent of the kinases screened that bind with a PoC value <10. A lower $S_{10}$ score indicates enhanced kinome-wide selectivity as a lower percentage of kinases bind with high affinity (PoC values closer to zero). Binding to a single kinase would generate an $S_{10}$ score of 0.002, while binding to 10 kinases yields an $S_{10}$ score of 0.025. WT human kinases in the *scan*MAX panel with PoC ≤ 20 (and GSK3β) for **B1**, with PoC < 10 for **B4**, and PoC < 35 for **B12** are also included in *Figure 3*. The kinome tree diagrams in *Figure 3* were created based on WT kinases with PoC < 10 for each compound.

## Biochemical assays

A homogeneous competition binding assay for CDKL5 was executed at Luceome Biotechnologies, LLC (Tuscon, AZ) using their *KinaseSeeker* technology (*Jester et al., 2010*). Briefly, this is a luminescence-based assay that relies on displacement of an active site-dependent probe by an ATP-competitive test compound. A change in luminescence signal indicates binding. These assays are robust and highly sensitive with low background and minimal interference from test compounds. Test compounds were screened at 11 different concentrations in duplicate and percentage of activity remaining plotting against compound concentration to calculate the corresponding $IC_{50}$ value. Compounds **B1**, **B4**, and **B12** were evaluated in dose–response (12-pt curve) format in duplicate. Curves with error bars showing standard deviation are included in *Figure 3—figure supplements 1 and 2*, and $IC_{50}$ values generated are embedded in tables within *Figure 3*.

Radiometric enzymatic assays were executed at Eurofins DiscoverX Corporation (Fremont) at the $K_m$ value for ATP to generate dose–response (9-pt) curves for all kinases listed in *Figure 3* and *Figure 3— figure supplements 1 and 2* (except for CDKL5). A detailed protocol for these assays, which includes

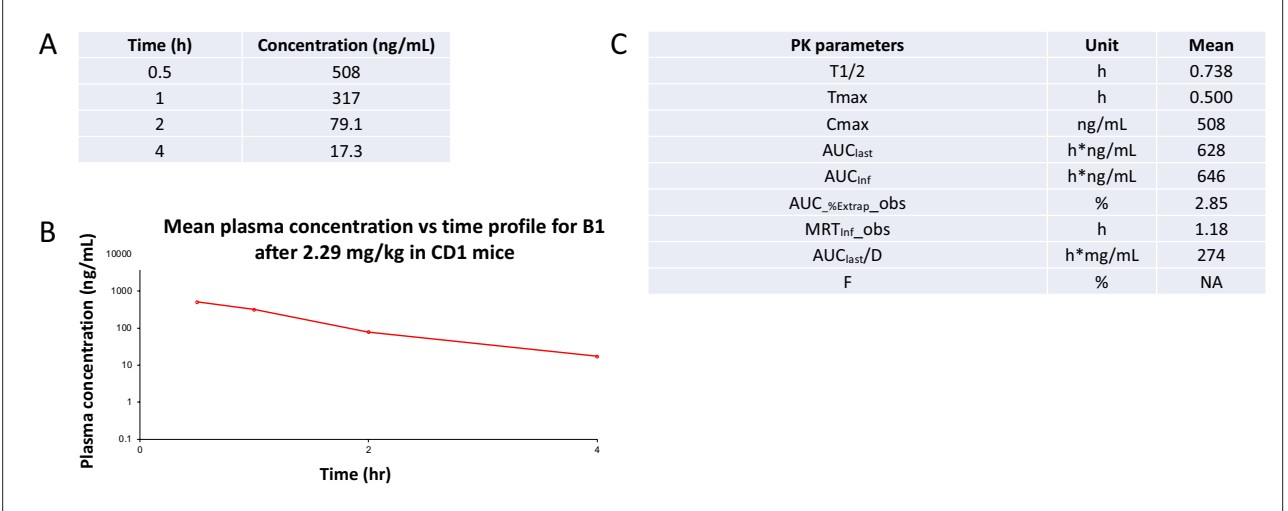

**Figure 12.** Snapshot of pharmacokinetic (PK) results for **B1**. (**A**) Plasma concentration measurements at timepoints post-administration. (**B**) Plot of mean plasma concentration over the time course of the PK study. (**C**) Summary of **B1** PK properties (n = 2). Values are expressed as the mean calculated after two animals were dosed. T1/2, half-life; Tmax, time of maximum observed concentration; Cmax, maximum observed concentration, occurring at Tmax (if not unique, then the first maximum is used); AUClast, area under the curve from the time of dosing to the last measurable positive concentration; AUCinf, area under the curve from dosing time extrapolated to infinity, based on the last observed concentration; AUC_%Extrap_obs, percentage of AUCinf due to extrapolation from last time point to infinity; MRTinf_obs, mean residence time extrapolated to infinity for a substance administered by intravascular dosing using observed last concentration; AUClast/D, area under the curve from the time of dosing to the last measurable concentration divided by the dose; *F*, bioavailability.

the protein constructs, substrate, and controls employed, is part of the Eurofins website: https://www.eurofinsdiscoveryservices.com. Briefly, either full-length or a truncate of a human kinase was incubated with [gamma-33P]-ATP in the presence of buffer, salts (magnesium source), and kinase-specific additives (such as cyclins for CDKs), where necessary, in the presence of a known substrate. Scintillation counting after quenching and washing with phosphoric acid allowed plotting of the data versus concentration and calculation of an $IC_{50}$ value. Inclusion of a positive control ensured assay performance.

## Kinetic solubility

The kinetic solubility of compound **B1** was evaluated using an aliquot of a 10 mM DMSO stock solution dissolved in phosphate-buffered saline solution (PBS) at pH 7.4. This analysis was done by Analiza, Inc (Cleveland, OH) as described previously (*Drewry et al., 2022a*). The reported solubility values in *Table 2* have been corrected for background nitrogen present in DMSO and the media.

## Microsomal stability

Mouse liver microsomal stability of compound **B1** was evaluated by Analiza, Inc as described previously (*Drewry et al., 2022b*). The assay was run in the presence and absence of NADPH. NADPH is required for the cytochrome P450 enzyme system, which is a common metabolism pathway of compounds. Any metabolism that occurs in the absence of NADPH would be due to non-NADPH-dependent enzymatic degradation and would be indicative of alternative metabolic pathways contributing to compound breaking down. Monoamine oxidase and carboxylesterases are examples of non-NADPH-dependent enzymes. This data generated by Analiza is included in *Table 2*.

## CDKL family thermal shift assays

Kinase domains of human CDKL1, CDKL2, CDKL3, and CDKL5 proteins were produced using the construct boundaries previously used to generate the X-ray crystal structures with PDB codes 4AGU, 4AAA, 3ZDU, and 4BGQ, respectively. 4 µM of the kinase domain of CDKL1, CDKL2, CDKL3, or CDLK5 in 10 mM HEPES-NaOH pH 7.4 and 500 mM NaCl was incubated with **B1** (12.5, 25, or 50 µM) in the presence of 5× SyPRO orange dye (Invitrogen). Next, fluorescence was evaluated using a Real-Time PCR Mx3005p machine (Stratagene). A previously reported protocol was followed to run the Tm

**Figure 13.** Scheme 1: preparation of compound **B1**. (*a*) EtOH, NaBH$_4$, acetone, 80°C, 1 hr, 57% yield; (*b*) HATU, DIPEA, 1-(*tert*-butoxycarbonyl) piperidine-4-carboxylic acid, DMF; (*c*) TFA, CH$_2$Cl$_2$, 36% yield over two steps. *5-(((5-isopropyloxazol-2-yl)methyl)thio)thiazol-2-amine (Int 1)*. To a flask was added 5-thiocyanatothiazol-2-amine (400 mg, 1.0 equiv, 2.5 mmol) in EtOH (24 mL), followed by addition of NaBH$_4$ (183 mg, 1.9 equiv, 4.8 mmol) portion-wise at room temperature. The mixture was stirred for 1 hr at room temperature then acetone (12 mL) was added. After 1 hr, a solution of 2-(chloromethyl)–5-isopropyloxazole (412 mg, 1.0 equiv, 2.6 mmol) in EtOH (4 mL) was added, and the reaction was heated at 80°C for 1 hr. The resulting mixture was cooled, concentrated in vacuo, and then partitioned between EtOAc and brine. The organic phase was separated, dried with MgSO$_4$, and concentrated in vacuo to give a crude solid. The crude material was triturated with diethyl ether/hexane to provide the desired product 4-(((5-isopropyloxazol-2-yl)methyl)thio)thiazol-2-amine as a red oil (368 mg, 57 %). *N-(5-(((5-isopropyloxazol-2-yl)methyl)thio)thiazol-2-yl)piperidine-4-carboxamide 2,2,2-trifluoroacetate* (**B1**). To a flask was added 1-(*tert*-butoxycarbonyl)piperidine-4-carboxylic acid (99 mg, 1.1 equiv, 431 μmol) and DIPEA (205 μL, 3 equiv, 1.2 mmol) and HATU (194 mg, 1.3 equiv, 509 μmol) and the flask was stirred at room temperature for approximately 15 min. Next, 5-(((5-isopropyloxazol-2-yl)methyl)thio)thiazol-2-amine (100 mg, 1.0 equiv, 392 μmol) was added and the reaction was stirred at room temperature for 16 hr. The reaction mixture was concentrated and purified by column chromatography (SiO$_2$, MeOH 0–10% in CH$_2$Cl$_2$) to yield the desired product as a white solid to yield the intermediate *tert*-butyl 4-((5-(((5-isopropyloxazol-2-yl)methyl)thio)thiazol-2-yl)carbamoyl)piperidine-1-carboxylate. To this intermediate was added 20% TFA and CH$_2$Cl$_2$ (5 mL) and the reaction stirred for 1 hr. The reaction was concentrated in vacuo and after addition of MeOH a white precipitate crashed out. The solid was filtered under vacuum to yield the desired product *N-(5-(((5-isopropyloxazol-2-yl)methyl)thio) thiazol-2-yl)piperidine-4-carboxamide 2,2,2-trifluoroacetate* as a white solid (20 mg). The remaining crude material was purified by preparative HPLC (MeOH 10–100% in H$_2$O [+0.05% TFA]) to yield the desired product *N-(5-(((5-isopropyloxazol-2-yl)methyl)thio)thiazol-2-yl)piperidine-4-carboxamide 2,2,2-trifluoroacetate* (**B1**) as a white solid (32 mg). Yield over two steps *N-(5-(((5-isopropyloxazol-2-yl)methyl)thio)thiazol-2-yl)piperidine-4-carboxamide 2,2,2-trifluoroacetate* (52 mg, 36%).

The online version of this article includes the following figure supplement(s) for figure 13:

**Figure supplement 1.** Characterization of *N-(5-(((5-isopropyloxazol-2-yl)methyl)thio)thiazol-2-yl)piperidine-4-carboxamide 2,2,2-trifluoroacetate* (**B1**).

shift assays and assess melting temperatures (*Fedorov et al., 2007*). Data is included in *Figure 3— figure supplement 2*. Error bars represent SD.

## CDKL5 enzymatic assays

These assays were executed as previously described (*Lin et al., 2005*; *Kim et al., 2020b*). Briefly, full-length FLAG-tagged WT (transcript variant II, NCBI Reference Sequence: NM_001037343.2) or kinase dead (KD, CDKL5 K42R) constructs of full-length human CDKL5 were subcloned into a pT7CFE1-CHis plasmid (Thermo Fisher). A HeLa cell lysate-based Kit (1-Step Human Coupled IVT Kit—DNA, 88881, Life Technologies) enabled in vitro translation of these constructs. His Pur cobalt spin columns (Thermo Scientific) were used to purify the in vitro-translated proteins. To execute the in vitro kinase assays, myelin basic protein (Active Motif, 31314) was employed as a substrate for recombinant CDKL5. Myelin basic protein is used as a substrate for multiple kinases, both serine/threonine and tyrosine kinases, to enable in vitro kinase assays due to the presence of multiple sites for phosphorylation. As such, we and others have used this protein as a kinase substrate for evaluating kinase activity (*Kim et al., 2020b*; *Kameshita et al., 2014*). These two components were incubated in kinase buffer (Cell Signaling, 9802) supplemented with or without ATP (50 μM) for 30 min at 30°C, followed by kinase assays run using the ADP-Glo Kinase Assay kit (Promega). After the termination of kinase assay, NuPAGE LDS Sample Buffer (4×) was added to the reaction mixture and the samples were heated at 100°C for 10 min. The protein samples were run on Invitrogen Bis-tris gradient midi-gels and transferred to PVDF membrane for western blot analysis. Primary antibody used for western blot analysis was from Cell Signaling: FLAG (#14793) and was used at 1:1,000 dilution. Anti-rabbit HRP-conjugated secondary antibody was from Jackson ImmunoResearch (#111-035-144) and used at 1:5000 dilution. Immunoblot signals were detected using SignalFire ECL Reagent (#6883) and X-ray films (Bioland #A03-02). Canon LiDE400 scanner was used for scanning the films. Precision Plus Protein Dual Color Standard prestained protein marker (Bio-Rad) was used to estimate the molecular weight of sample proteins.

**Figure 14.** Scheme 2: preparation of compounds **B4** and **B12**. (*a*) Propylphosphonic anhydride (T3P), DIPEA, THF, 0°C, 1 hr, 93% yield; (*b*) palladium on carbon (Pd/C), MeOH, THF, 25°C, 16 hr, 99% yield; (*c*) 4-methoxybenzoic acid, HATU, DIPEA, THF, 25°C, 17 hr, 72% yield; (*d*) HCl, dioxane, 25°C, 1 hr, 58% yield; (*e*) 3-cyanobenzoic acid, T3P, DIPEA, THF, 25°C, 16 hr; (*f*) HCl, dioxane, 25°C, 1 hr, 39% yield over two steps. General procedure for preparation of B4 and B12: To a flask was added 4-nitro-1*H*-pyrazole-3-carboxylic acid (1.00 g, 1.0 equiv, 6.4 mmol), 1-Boc-4-aminopiperidine (1.28 g, 1.0 equiv, 6.4 mmol), and DIPEA (4.21 mL, 4.0 equiv, 25.5 mmol) in THF (20 mL). The reaction mixture was cooled to 0°C, and then T3P (6.08 mL, 1.5 equiv, 9.5 mmol) was slowly added. After stirring for 1 hr at 0°C, the reaction mixture was concentrated in vacuo, and then partitioned between EtOAc and brine. The organic phase was separated, dried with MgSO$_4$, and concentrated in vacuo to give a crude solid. The crude material was purified using an automated purification system (SiO$_2$) 90%/10% hexanes/EtOAc to 100% EtOAc to give *tert*-butyl 4-(4-nitro-1*H*-pyrazole-3-carboxamido)piperidine-1-carboxylate (**Int 2**) as an amorphous solid (2.00 g, 93% yield). Palladium on carbon (5%, 31.3 mg, 0.1 equiv, 0.295 mmol) was added to *tert*-butyl 4-(4-nitro-1*H*-pyrazole-3-carboxamido)piperidine-1-carboxylate (**Int 2**, 1.0 g, 1.0 equiv, 2.95 mmol) in a mixture of methanol (10 mL) and THF (3.0 mL). The solution was placed under an atmosphere of H$_2$ at room temperature. After 16 hr, the solution was filtered through SiO$_2$ and the eluent was concentrated to afford *tert*-butyl 4-(4-amino-1*H*-pyrazole-3-carboxamido)piperidine-1-carboxylate (**Int 3**) as an amorphous solid (900 mg, 99% yield). To a flask was added *tert*-butyl 4-(4-amino-1*H*-pyrazole-3-carboxamido)piperidine-1-carboxylate (**Int 3**, 1 equiv) and the corresponding carboxylic acid (1.2 equiv) and DIPEA (4.0 equiv) in THF (0.5 M) at room temperature. The reaction mixture was then treated slowly with T3P (3.0 equiv). After stirring for 16–17 hr, the reaction mixture was concentrated in vacuo, and then partitioned between EtOAc and brine. The organic phase was separated, dried with MgSO$_4$, and concentrated in vacuo to give a crude solid. The crude material was purified using an automated purification system (SiO$_2$) 90%/100% hexanes/EtOAc to 100% EtOAc to give an intermediate, which was stirred in HCl/dioxane (3.3 mL) for 1 hr at room temperature. Solvent was removed and the crude material was purified using an automated purification system (SiO$_2$) 90%/100% hexanes/EtOAc to 100% EtOAc to give the desired product (**B4** or **B12**) as an amorphous solid (39–42% yield over two steps).

The online version of this article includes the following figure supplement(s) for figure 14:

**Figure supplement 1.** Characterization of 4-(4-methoxybenzamido)-*N*-(piperidin-4-yl)–1H-pyrazole-3-carboxamide (**B4**).

**Figure supplement 2.** Characterization of 4-(3-cyanobenzamido)-*N*-(piperidin-4-yl)–1H-pyrazole-3-carboxamide (**B12**).

## Snapshot PK study

The pharmacokinetics of compound **B1** was evaluated by Pharmaron (San Diego, CA) following a single intraperitoneal administration to CD1 mice (two males). A dose of 2.29 mg/kg of the TFA salt of compound **B1** in NMP/Solutol/PEG-400/normal saline (v/v/v/v, 10:5:30:55) was prepared just before use. This dose was calibrated due to the compound being a salt form. Plasma was sampled 0.5, 1, 2, and 4 hr post-dose. Mean values from the two animals without dispersion are shown in *Figure 12*. Tmax (*Figure 12*) was defaulted to the earliest time point and likely occurred earlier than 0.5 hr. Similarly, Cmax (*Figure 12*) was recorded as the concentration after 0.5 hr, the first sampling time point. Bioanalytical assays were run using Prominence HPLC and AB Sciex Triple Quan 5500 LC/MS/MS instruments, and a HALO column (90A, C18, 2.7 μm, 2.1 × 50 mm). Snapshot PK results are included in *Figure 12*. PK parameters were estimated by non-compartmental model using WinNonlin 8.3 (Certara, Princeton, NJ). The lower limit of quantification for plasma sampling was 1.53 ng/mL.

## Analysis of brain/plasma concentration

The brain/plasma concentration of compound **B1** was evaluated by Pharmaron following a single intraperitoneal administration to CD1 mice (three males per dose). Two doses (2.29 mg/kg and 7.63 mg/

kg) of the TFA salt of compound **B1** in NMP/Solutol/PEG-400/normal saline (v/v/v/v, 10:5:30:55) were prepared just before use. These doses were calibrated due to the compound being a salt form. The same dose as was used for Snapshot PK (2.29 mg/kg) was chosen to have a more comprehensive profile, including brain penetrance, at this concentration. Since brain penetration was anticipated to be low, a higher concentration was also selected. Plasma and brain samples were collected 1 hr post-dose. Brain samples were homogenized at a ratio of 1:3 with PBS (W/V, 1:3) and then final brain concentrations corrected. Bioanalytical assays were run using Prominence HPLC and AB Sciex Triple Quan 5500 LC/MS/MS instruments, and a HALO column (90A, C18, 2.7 µm, 2.1 × 50 mm). Brain/plasma concentration results are included in *Table 3*. The lower limit of quantification for plasma sampling was 1.53 ng/mL and 0.763 ng/mL for brain.

## Animals: The Francis Crick Institute (TFCI)

Animals were housed in a controlled environment with 12 hr light/dark cycle. They were fed ad libitum and used in accordance with the Animals (Scientific Procedures) Act 1986 of the United Kingdom. Rat handling and housing was performed according to the regulations of the Animal (Scientific Procedures) Act 1986. Animal studies were approved by the Francis Crick Institute ethical committee and performed under U.K. Home Office project license (PPL P5E6B5A4B).

## Rat neuron primary culture: TFCI

Primary cortical cultures were prepared from embryonic day (E) 18.5 embryos of Long–Evans rats as described (*Roşianu et al., 2023*). Pregnant females were culled using cervical dislocation, embryos were removed from the uterus, and the brains were taken out. Cortices were dissected out, pooled from multiple animals, and washed three times with HBSS. An incubation with 0.25% trypsin for 15 min at 37°C was followed by four times washing with HBSS. Cells were dissociated and then counted using a hemocytometer. Neurons were plated on 12-well culture plates at a density of 300,000 cells per well. The wells were coated with 0.1 M borate buffer containing 60 µg/mL poly-D-lysine and 2.5 µg/mL laminin and placed in the incubator overnight. Neurons were plated with minimum essential medium (MEM) containing 10% FBS, 0.5% dextrose, 0.11 mg/mL sodium pyruvate, 2 mM glutamine, and penicillin/streptomycin. After 4 hr, cultures were transferred to neurobasal medium containing 1 mL of B27 (Gibco), 0.5 mM GlutaMAX, 0.5 mM glutamine, 12.5 µM glutamate, and penicillin/streptomycin. Primary neuronal cultures were kept at 37°C and 5% $CO_2$. Every 3–4 d, 20–30% of the maintenance media was refreshed. Between DIV14-16, neurons were treated with 5 nM, 50 nM, 500 nM, and 5 µM of the different compounds for 1 hr. The compounds were added directly to the media and the plates were placed at 37°C for the time of the treatment. DMSO was added to the well for the control condition.

## Western blotting: TFCI

After treatment, neuronal cultures were lysed in 300 µL of 1× sample buffer (Invitrogen) containing 0.1 M DTT. Lysates were sonicated briefly twice and denatured at 70°C for 10 min. The samples were centrifuged at 13,300 rpm for 10 min and ran on NuPage 4–12% Bis-Tris polyacrylamide gels (Invitrogen). Proteins were transferred onto a Immobilon PVDF membrane (Millipore), which was then blocked in 4% milk in Tris-buffered saline containing 0.1% Tween-20 (TBST) for 30 min. Primary antibodies were incubated at 4°C overnight, and HRP-conjugated secondary antibodies at room temperature (RT) for 2 hr. The following primary antibodies were used: rabbit anti-CDKL5 (1:1000; Atlas HPA002847), rabbit anti-pS222 EB2 (1:2000; Covalab, from *Baltussen et al., 2018*), rat anti-EB2 (1:2000; Abcam ab45767), mouse anti-tubulin (1:100,000; Sigma T9026), rabbit anti-β-catenin (1:1000; Cell Signaling 9562), and rabbit anti phospho-β-catenin (Ser33/37 Thr41) (1:250, Cell Signaling 9561). The following secondary antibodies were used at a concentration of 1:10,000: HRP-conjugated anti-rabbit (Jackson 711-035-152), HRP-conjugated anti-mouse (Jackson 715-035-151), and HRP-conjugated anti-rat (Jackson 712-035-153). The membrane was developed using ECL reagent (Cytiva) and was visualized with an Amersham Imager 600 (GE Healthcare). Quantification of western blots was manually performed using Image Studio Lite Software (version 5.2). EB2 phosphorylation was measured relative to total EB2 and catenin phosphorylation to total catenin. The other proteins were normalized to tubulin, if not indicated otherwise.

## Animals: University of Colorado School of Medicine (UC-SOM)

All studies conformed to the requirements of the National Institutes of Health *Guide for the Care and Use of Laboratory Rats* and were approved by the Institutional Animal Care and Use subcommittee of the University of Colorado Anschutz Medical Campus (protocol 00411). Timed-pregnant Sprague–Dawley rats (Charles Rivers Labs, Wilmington, MA) gave birth in-house. All rodents were housed in micro-isolator cages with water and chow available ad libitum.

## Hippocampal slice preparation and electrophysiology: UC-SOM

As done previously (*Cornejo et al., 2007*; *Bernard et al., 2014*; *Bernard et al., 2013*), following rapid decapitation and removal of the rat brain at postnatal day (P) 20–30, sagittal hippocampal slices (400 µm) were made using a vibratome (Leica VT 1200, Buffalo Grove, IL) in ice-cold sucrose artificial cerebral spinal fluid (sACSF: 206 mM sucrose, 2.8 mM KCl, 1 mM $CaCl_2$, 3 mM $MgSO_4$, 1.25 mM $NaH_2PO_4$, 26 mM $NaHCO_3$, 10 mM D-glucose, and bubbled with 95%/5% $O_2/CO_2$). Following removal of CA3, slices were recovered in a submersion-type chamber perfused with oxygenated artificial cerebral spinal fluid (ACSF: 124 mM NaCl, 3 mM KCl, 1 mM $MgSO_4$, 2 mM $CaCl_2$, 1.2 mM $NaH_2PO_4$, 26 mM $NaHCO3$, 10 mM D-glucose and bubbled with 95%/5% $O_2/CO_2$) at 28°C for at least 90 min and then submerged in a recording chamber perfused with recirculated ACSF. All electrophysiology was performed in the CA1 region at 28°C. Drugs were added to ACSF. Two twisted-tungsten bipolar-stimulating electrodes were offset in the CA1 to stimulate one or two independent Schaffer collateral-commissural pathways using a constant current source (WPI, Sarasota, FL) with a fixed duration (20 µs), each at a rate of 0.033 Hz. fEPSPs were recorded from the stratum radiatum (or stratum pyramidale region where indicated) region of CA1 using a borosilicate glass (WPI) microelectrode (pulled [Sutter, Novato, CA] to 6–9 MΩ when filled with ACSF), amplified 1000× (WPI and Warner, Hamden, CT), and digitized (National Instruments, Austin, TX) at 20 kHz using winLTP-version 2.4 (*Anderson and Collingridge, 2001*) to follow fEPSP slope (averaged over four EPSPs), measured using 20–80% rise times, expressed as percent of baseline, during the course of an experiment. To be sure only 'healthy' slices were included in our studies, responses had to meet several criteria: fiber volleys less than 1/3 of response amplitude and peak responses larger than 0.5 mV; responses and fiber volley must be stable (<5% drift). Following baseline stabilization of fEPSP slope at ~50% of maximal slope for at least 30 min, measurements of synaptic physiology were performed. Stimulation was adjusted to make 4–6 measurements from ~10 to 90% maximal slope to measure input-output (I/O) curves; recordings that deviated from linear slope were rejected. Paired-pulse ratios of slopes and peaks were measured at ~50% of maximal slope. For measurements of NMDA-receptor-dependent population spikes (*Coan and Collingridge, 1985*), $Mg^{2+}$ was omitted from recording ACSF, the recording electrode was placed in stratum pyramidale, stimulation was adjusted to evoke a first population spike >2 mV, and stabilization of the second NMDA-receptor-dependent population spike (measured from the coastline of the fEPSP) for >30 min was allowed prior to measuring the effect of antagonists. Theta burst LTP was induced at test strength of 2 trains, separated by 20 s, of 10 trains of 4 stimuli at 100 Hz, separated by 200 ms. Potentiation was statistically compared over the last 5 min.

For whole-cell, voltage-clamp electrophysiological recordings of excitatory postsynaptic currents (EPSCs), 350 µm horizontal hippocampal slices were prepared from rats at P 20–30 similar to previously (*Sanderson et al., 2021*; *Sanderson et al., 2016*). After 15 min recovery at 34°C, slices were maintained at RT until recording in modified ACSF containing (in mM) 126 NaCl, 5 KCl, 2 $CaCl_2$, 1.25 $NaH_2PO_4$, 1 $MgSO_4$, 26 $NaHCO_3$, 10 glucose, and 2 N-acetyl cysteine. Slices were incubated in either CAF-382 (**B1**) (100 nM) or vehicle (water) in modified ACSF for at least 1 hr prior to recording. Using infrared–differential interference contrast microscopy for visualization, whole-cell recordings (Axopatch 200B amplifier and pClamp software [Molecular Devices]) from CA1 pyramidal neurons (series resistance between 3 and 6 MΩ) were performed at 29–30°C with an intracellular solution containing (in mM) 130 Cs gluconate, 1 CsCl, 1 $MgSO_4$, 10 HEPES, 1 EGTA, 4 MgATP, 0.4 MgGTP, spermine (0.01), and 2 QX-314, pH 7.3. AMPA-type EPSCs, evoked using a bipolar tungsten-stimulating electrode as above, at a holding potential of −65 mV were pharmacologically isolated using picrotoxin (50 µM; Tocris). EPSC peaks were evoked every 20 s. After establishing whole-cell access, a 3 min baseline recording at −65 mV was followed by LTP induction using a pairing protocol: cells were depolarized to 0 mV for 90 s and stimulated every 20 s then followed by delivery of 1 s × 100 Hz high-frequency

stimulation. Evoked EPSC amplitudes were subsequently monitored at –65 mV. Potentiation was statistically compared over the last 5 min.

## Western blotting: UC-SOM

Hippocampal slices were prepared and recovered as for electrophysiology with the addition of cuts to isolate CA1 from the remaining hippocampus; after recovery, the slices from a given rat were incubated in inhibitor or vehicle (water) in ACSF for 2 hr. Following this, slices were suspended in STE buffer, sonicated, boiled for 5 min, and then frozen until further use. To minimize the effects of slice preparation (*Osterweil et al., 2010*) and control for slice quality, only slices that came from a preparation that met electrophysiological criteria were used. All concentrations were quantified with BCA and then loaded in duplicate on 10% polyacrylamide gel and a five-point dilution series of naive rat hippocampal homogenate was included on each gel as a standard curve for quantification of immunoreactivity/μg loaded protein (*Grosshans and Browning, 2001*). Following transfer to PolyScreen PVDF transfer membrane (Genie, Idea Scientific Company, Minneapolis, MN), blots were blocked in BSA or Carnation nonfat dry milk for 1 hr and incubated either 1 hr at RT or overnight at 4°C with antibodies. The following primary antibodies were used: sheep anti-CDKL5 (1:2000; University of Dundee, anti-CDKL5 [350-650]), rabbit anti-pS222 EB2 (1:1000; Covalab, from *Baltussen et al., 2018*), and rat anti-EB2 (1:1000; Abcam ab45767). Blots were then subjected to three 10 min washings in Tris-buffered saline (140 mm NaCl, 20 mm Tris pH 7.6) plus 0.1% Tween 20 (TTBS), before being incubated with anti-sheep, anti-rat, or anti-rabbit secondary antibody (1:5000) in 1% BSA or milk for 1 hr at RT, followed by three additional 10 min washes preformed with TTBS. Immunodetection was accomplished using a chemiluminescent substrate kit (SuperSignal West Femto Maximum Sensitivity Substrate; Pierce) and the Alpha Innotech (Alpha Innotech, San Leandro, CA) imaging system. Quantification was performed using AlphaEase software (Alpha Innotech) and Excel (Microsoft, Redmond, WA). Immunoreactivity was reported as the density of sample bands relative to the standard curve. For phospho-proteins, ratios of immunoreactivity/μg to totals are reported without standardization. Only values falling within the standard curve generated from the dilution series included on each gel were incorporated into the final analysis. Some of the blots were then stripped (Restore PLUS Western Blot Stripping Buffer; Thermo Scientific, Rockford, IL) and reblotted if needed.

## Statistical analysis: OSU

Data were analyzed via GraphPad Prism 9. Each assay was run in triplicate, and mean values are graphed in *Figure 4* with error bars showing standard deviation (SD). In *Figure 4*, statistical analysis was done using one-way ANOVA with Dunnett's multiple-comparisons test. Thresholds for significance (p-values) were set at \*\*\*p<0.0001 and nonsignificant statistics were not included.

## Statistical analysis: TFCI

Data were analyzed using GraphPad Prism 9. Exact values of n and statistical methods are mentioned in the figure legends. Each concentration was compared to the control using a one-way ANOVA test. A p-value >0.05 was not considered statistically significant. Thresholds for significance were placed at \*p≤0.05, \*\*p≤0.01, \*\*\*p≤0.001, and \*\*\*\*p≤0.0001. All errors bars in the figures are SD where indicated. Nonsignificant statistics were not included.

## Statistical analysis: UC-SOM

Data are expressed as mean ± SEM with n = number of rats for a given treatment, unless otherwise stated. Data are plotted using SigmaPlot 12.5 (Systat, Chicago, IL). Mann–Whitney rank-sum, one-way, two-way, and repeated measures (RM) ANOVA (Holm–Sidak post hoc testing) and Student's *t*-tests were used, where indicated and appropriate, for statistical comparisons for electrophysiological and biochemical using SigmaPlot 12.5 (Systat). Significance is reported at p<0.05 unless otherwise stated. All errors bars in the figures are standard error (SE) where indicated. Nonsignificant statistics were not included.

## Acknowledgements

NanoBRET constructs for CDKL5, GSK3α, and GSK3β were kindly provided to the SGC-UNC team by Promega. TREE*spot* kinase interaction mapping software (http://treespot.discoverx.com) was

used to prepare the kinome trees in *Figure 3*. The authors thank the Diamond Light Source for beamtime (proposal mx28172) as well as the staff of beamline i04 for their guidance. Funding was provided by NIH-NINDS NS112770 (all) and Ponzio Family Chair in Neurology Research (AC, TAB), and The Structural Genomics Consortium (SGC) is a registered charity (number 1097737) that receives funds from Bayer AG, Boehringer Ingelheim, the Canada Foundation for Innovation, Eshelman Institute for Innovation, Genentech, Genome Canada through Ontario Genomics Institute [OGI-196], EU/EFPIA/OICR/McGill/KTH/Diamond, Innovative Medicines Initiative 2 Joint Undertaking [EUbOPEN grant 875510], Janssen, Merck KGaA (aka EMD in Canada and USA), Pfizer, the São Paulo Research Foundation-FAPESP, and Takeda (CIW, CAF, HWO, YL, FMB, JLS, IMG, DHD, ADA). Research reported in this publication was supported in part by the NC Biotechnology Center Institutional Support Grant 2018-IDG-1030 (DHD, ADA), NIH U24DK116204 (DHD, ADA), and NIH 1R44TR001916 (DHD, ADA). This work was supported by the Francis Crick Institute which receives its core funding from Cancer Research UK (CC2037), the UK Medical Research Council (CC2037), and the Wellcome Trust (CC2037); Loulou Foundation Project Grant (11015). Margaux Silvestre is a recipient of the Loulou foundation junior fellowship award (2021). For the purpose of Open Access, the author has applied a CC BY public copyright license to any Author Accepted Manuscript version arising from this submission.

## Additional information

### Competing interests

Tim A Benke: Consultancy for AveXis, Ovid, GW Pharmaceuticals, International Rett Syndrome Foundation, Takeda, Taysha, CureGRIN, GRIN Therapeutics, Alcyone, Neurogene, and Marinus; Clinical Trials with Acadia, Ovid, GW Pharmaceuticals, Marinus and RSRT; all remuneration has been made to his department. Alison D Axtman: Advisor for Proteic Bioscience Inc. The other authors declare that no competing interests exist.

### Funding

| Funder | Grant reference number | Author |
|---|---|---|
| National Institute of Neurological Disorders and Stroke | NS112770 | Anna Castano |
| Children's Hospital Colorado Foundation | Ponzio Family Chair in Neuroscience Research | Anna Castano |
| Structural Genomics Consortium | | Carrow I Wells |
| NC Biotechnology Center Institutional Support Grant | 2018-IDG-1030 | David H Drewry |
| National Institutes of Health | U24DK116204 | David H Drewry |
| National Institutes of Health | R44TR001916 | David H Drewry |
| Cancer Research UK | CC2037 | Margaux Silvestre |
| Medical Research Council | CC2037 | Margaux Silvestre |
| Wellcome Trust | CC2037 | Margaux Silvestre |
| LouLou Foundation | 11015 | Margaux Silvestre |

The funders had no role in study design, data collection and interpretation, or the decision to submit the work for publication. For the purpose of Open Access, the authors have applied a CC BY public copyright license to any Author Accepted Manuscript version arising from this submission.

## Author contributions

Anna Castano, Data curation, Formal analysis, Writing – review and editing; Margaux Silvestre, Data curation, Formal analysis, Funding acquisition, Methodology, Writing – review and editing; Carrow I Wells, Jennifer L Sanderson, Han Wee Ong, William Richardson, Data curation, Formal analysis, Methodology, Writing – review and editing; Carla A Ferrer, Josie A Silvaroli, Frances M Bashore, Jeffery L Smith, Isabelle M Genereux, Kelvin Dempster, Navlot S Pabla, Alex N Bullock, Data curation, Formal analysis, Investigation, Methodology, Writing – review and editing; Yi Lang, Data curation, Formal analysis, Investigation, Writing – review and editing; David H Drewry, Formal analysis, Funding acquisition, Investigation, Methodology, Writing – review and editing; Tim A Benke, Conceptualization, Resources, Data curation, Formal analysis, Supervision, Funding acquisition, Investigation, Visualization, Methodology, Writing – original draft, Project administration, Writing – review and editing; Sila K Ultanir, Conceptualization, Data curation, Formal analysis, Supervision, Funding acquisition, Investigation, Visualization, Methodology, Writing – original draft, Project administration, Writing – review and editing; Alison D Axtman, Conceptualization, Resources, Data curation, Formal analysis, Supervision, Funding acquisition, Validation, Investigation, Visualization, Methodology, Writing – original draft, Project administration, Writing – review and editing

## Author ORCIDs

Margaux Silvestre (ID) https://orcid.org/0000-0003-1377-477X
Carrow I Wells (ID) https://orcid.org/0000-0003-4799-6792
Frances M Bashore (ID) https://orcid.org/0000-0003-4241-9873
Jeffery L Smith (ID) https://orcid.org/0000-0003-2189-0420
Kelvin Dempster (ID) https://orcid.org/0009-0009-0750-0175
Navlot S Pabla (ID) https://orcid.org/0000-0001-9408-0539
Tim A Benke (ID) https://orcid.org/0000-0002-6969-5061
Sila K Ultanir (ID) https://orcid.org/0000-0001-5745-3957

## Ethics

The Francis Crick Institute (TFCI). Rat handling and housing was performed according to the regulations of the Animal (Scientific Procedures) Act 1986. Animal studies were approved by the Francis Crick Institute ethical committee and performed under U.K. Home Office project license (PPL P5E6B5A4B). Animals-University of Colorado School of Medicine (UC-SOM). All studies conformed to the requirements of the National Institutes of Health Guide for the Care and Use of Laboratory Rats and were approved by the Institutional Animal Care and Use subcommittee of the University of Colorado Anschutz Medical Campus (protocol 00411). Timed-pregnant Sprague–Dawley rats (Charles Rivers Labs, Wilmington, MA, USA) gave birth in-house. All rodents were housed in micro-isolator cages with water and chow available ad libitum.

## Decision letter and Author response

Decision letter https://doi.org/10.7554/eLife.88206.sa1
Author response https://doi.org/10.7554/eLife.88206.sa2

---

# Additional files

## Supplementary files

MDAR checklist

## Data availability

Materials and data availability statement has been provided in the text: Data are available via Harvard Dataverse (https://doi.org/10.7910/DVN/FRRMYI). New reagents are available from the senior authors.

The following dataset was generated:

| Author(s) | Year | Dataset title | Dataset URL | Database and Identifier |
|-----------|------|---------------|-------------|-------------------------|
| Benke TA | 2026 | Replication Data for: Discovery and characterization of a specific inhibitor of serine-threonine kinase cyclin dependent kinase-like 5 (CDKL5) demonstrates role in hippocampal CA1 physiology. | https://doi.org/10.7910/DVN/FRRMYI | Harvard Dataverse, 10.7910/DVN/FRRMYI |

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
