## [Editor Report]

This important study reports selective CDKL5 inhibitors that may be of high interest to investigate the role of this kinase in disease (particularly, in CDKL5 deficiency disorder) and to address unsolved issues of inconsistency in the phenotypic characterization of CDKL5-deficient knockout mice. The evidence supporting the conclusions is convincing, with rigorous biochemical, in vitro and ex vivo assays. The work will be of interest to cell and medical biologists and epileptologists working in the fields of epilepsy and neural excitation.

---

## [Decision Letter]

**Decision letter after peer review:**

Thank you for submitting your article "Discovery and characterization of a specific inhibitor of serine-threonine kinase cyclin dependent kinase-like 5 (CDKL5) demonstrates role in hippocampal CA1 physiology" for consideration by *eLife*. Your article has been reviewed by 3 peer reviewers, and the evaluation has been overseen by a Reviewing Editor and Amy Andreotti as the Senior Editor.

Essential revisions:

1) Please further describe the screening procedure that led to the reported selective inhibitors (e.g., size of the screened libraries, description of the binding and orthogonal assays, reasons to use myelin basic protein as a substrate for CDKL5 in the in vitro kinase assays, etc.).

2) Please discuss potential limitations to using the reported inhibitors based on the seemingly limited brain penetration.

3) Please revise figures, to clearly show statistically significant differences when detected, add legends, improve description in Figures' captions, etc.

4) On the basis of the reviewers' comments. Please further discuss the statement "At issue, is whether observations of CDKL5 activity, either through genetic manipulations or interference with the kinase, are specific and not potentially subsumed by the activity of other kinases. Given the reduction of phosphorylation of the specific substrate, EB2, this is less likely here".

*Reviewer #1 (Recommendations for the authors):*

In the introduction, when stating: "Epilepsy in CDD does not respond to any antiseizure medications[11]; a newly approved therapy provides modest improvements[12]", the identity of this newly approved therapy should be revealed.

Please indicate the source(s) of reactants used for synthesis.

Even if some of the reported data were obtained through an outsource service (e.g., the studies on kinome-wide selectivity) the details of the assay should be exhaustively described under the methods section to ensure reproducibility. Although a reference (ref. 43) has been included related to kinase inhibition selectivity, the authors are not from the same organization that has measured selectivity here, so I believe the details on the methodology should be made explicit.

Regarding the preliminary PK study, I am curious about how tmax was estimated at 0.5 hours considering this is the earliest sampling point. Also, abbreviations used in Figure S10 should be defined and data should be presented with dispersion. How was the administered dose chosen? I am not sure about reporting Cmax and tmax with such a limited number of data points and no earlier sampling times.

How were the doses used for the brain/plasma partitioning chosen?

Details on the housing and feeding of the laboratory animals at The Francis Crick Institute should be provided.

What is the total number of compounds included in the extensive libraries of SNS-032 and AT-7519 analogs?

Whereas the S10 score is defined in the Figure 3 caption, it would be a good idea to include its definition in the main text, possibly in the Method section.

Have the authors obtained data on the unbound fraction of B1?

I am not convinced by the argument that, since B1 reduces the phosphorylation of the specific substrate EB2, interference with other kinases expressed in the brain might not be at play here. Please elaborate further.

The authors indicate that brain penetration of B1 is rather low. Do they see this as an obstacle to using their inhibitor to further study the role of CDKL5 in CDD using in vivo assays? What about determining the kpuu, which is much more relevant pharmacologically speaking? (the shift in potency from biochemical to cell-based assays also suggests a permeability problem). Haven´t the authors considered a hit-to-lead campaign to improve B1 PK profile?

The potential utility of B1 could be further highlighted towards the concluding section of the manuscript.

I suggest indicating the statistically significant differences in the means in Figure 6 graphically. Can the authors explain the apparent lack of a clear dose-response behavior?

Seemingly, the authors have not observed their code to indicate different levels of statistical significance in Figure 7.

What conclusion(s) can the authors draw from the similar microsomal stability observed with and without NADPH? Are the differences between 0 and 30 min statistically significant? Please provide a wider interpretation of the results.

*Reviewer #2 (Recommendations for the authors):*

I have several questions to clarify some descriptions.

1. Selectivity score

Figure 3 and the corresponding description of the selectivity score – the meaning of these is hard to understand. Do the dots in kinome tree diagrams represent 403 kinases? Which dot is CDKL5? Why were red circles drawn only on the CMGC family? The authors described that the S10 score is a way to express selectivity that corresponds with the percent of the kinases screened that bind with a PoC value <10 and indicated 0.017, 0.02, and 0.01 for B1, B4, and B12 respectively. But it is hard to interpret the significance of these values. Can you provide some scale or perspective and let us know how selective these values are in terms of the kinome? Why do some kinases have two IC50 (nM) values?

2. Figure 7B

Please add legends for 3 traces.

3. Lines 653-656

"At issue, is whether observations of CDKL5 activity, either through genetic manipulations or interference with the kinase, are specific and not potentially subsumed by the activity of other kinases. Given the reduction of phosphorylation of the specific substrate, EB2, this is less likely here."

It is hard to understand the intent of these sentences.

*Reviewer #3 (Recommendations for the authors):*

The first part of the paper describing binding and orthogonal assays should be improved to render the read easier. For non-expert readers, it is impossible to really understand the specificity of the inhibitor, which is a central part of the manuscript. It is also not clear why the authors start with a panel of 25 inhibitors and test 20 in the rat neurons.

While the work in general is well presented it could be improved by being updated and, in a few cases, more precise.

Line 170: "epilepsy in CDD does not respond to any seizure medication" – that is a bit overstated.

Line 182: I suggest citing Terzic and colleagues' work from 2021 showing that CDKL5 is required also during adulthood.

Line 192-197: I suggest including De Rosa et al's work from 2023 suggesting a role of CDKL5 in GABAA-R mediated inhibition through the interaction with gephyrin.

Line 590: the work by Kontaxi et al. has been published.

Kinase assay: it is not clear whether full-length CDKL5, and which isoform, was used for the in vitro kinase assay. Furthermore, it is not clear why myelin basic protein was used as a substrate – does it contain the CDKL5 consensus site (which is by the way never mentioned in the paper)?

---

## [Author Response]

Essential revisions:1) Please further describe the screening procedure that led to the reported selective inhibitors (e.g., size of the screened libraries, description of the binding and orthogonal assays, reasons to use myelin basic protein as a substrate for CDKL5 in the in vitro kinase assays, etc.).

We have modified the Results to include the size of the screened libraries based around AT-7519 and SNS-032. Descriptions of the NanoBRET, Eurofins DiscoverX scanMAX, Luceome CDKL5 binding, and Eurofins radiometric assays were also added to the Methods and Results. We have updated Figures 3 and S6. In the Methods section, we have added more clarification about the use of myelin basic protein as a substrate for CDKL5 and pertinent references. These additional details have further delineated our screening procedure for the reader.

2) Please discuss potential limitations to using the reported inhibitors based on the seemingly limited brain penetration.

We have added sentences to the Discussion to further detail potential limitations to using the reported inhibitors and highlighting in which situations their use is highly recommended. Specifically, these inhibitors will be useful for in vitro studies to study the role of CDKL5 pathophysiology with some limited uses in vivo.

3) Please revise figures, to clearly show statistically significant differences when detected, add legends, improve description in Figures' captions, etc.

With attention to the reviewers’ specific comments, we have added additional text and revised the figures and legends to address these important issues. More detail about statistics were added to the captions of Figures 6, 7, S4–S6 and S10.

4) On the basis of the reviewers' comments. Please further discuss the statement "At issue, is whether observations of CDKL5 activity, either through genetic manipulations or interference with the kinase, are specific and not potentially subsumed by the activity of other kinases. Given the reduction of phosphorylation of the specific substrate, EB2, this is less likely here".

We agree with the reviewers that this is not clear. We have deleted these sentences and added further clarification to the discussion. The more comprehensive NanoBRET profiling of CAF382 (B1) supports that at the concentrations we used the results generated are specific to CDKL5 inhibition and not due to inhibition of off-target kinases.

Reviewer #1 (Recommendations for the authors):In the introduction, when stating: "Epilepsy in CDD does not respond to any antiseizure medications[11]; a newly approved therapy provides modest improvements[12]", the identity of this newly approved therapy should be revealed.

This was added to the introduction.

Please indicate the source(s) of reactants used for synthesis.

We have added the most common commercial vendors of reactants to the first section of Supplementary Information. We have completed the Key Reagent Table.

Even if some of the reported data were obtained through an outsource service (e.g., the studies on kinome-wide selectivity) the details of the assay should be exhaustively described under the methods section to ensure reproducibility. Although a reference (ref. 43) has been included related to kinase inhibition selectivity, the authors are not from the same organization that has measured selectivity here, so I believe the details on the methodology should be made explicit.

We have added more details to the Methods section for all assays, even those run externally. This should help ensure reproducibility.

Regarding the preliminary PK study, I am curious about how tmax was estimated at 0.5 hours considering this is the earliest sampling point. Also, abbreviations used in Figure S10 should be defined and data should be presented with dispersion. How was the administered dose chosen? I am not sure about reporting Cmax and tmax with such a limited number of data points and no earlier sampling times.

Tmax was defaulted to 0.5 hours because, as the reviewer stated, it was the earliest sampling point and also demonstrated the highest concentration (508 ng/mL after dosing at 2.29 mg/kg). All subsequent time points demonstrated concentrations of <320 ng/mL and decreased over time.

We have made a note about this Tmax value as well as the Cmax value in the corresponding Methods section. The abbreviations used in Figure S10 are now all defined in the legend. The data included in Figure S10 was provided by Pharmaron as mean values only, without dispersion. We have indicated this in the figure legend as well as in the corresponding Methods section. We were aiming for a dose of 5 mg/kg but then had to adjust slightly given that the compound was prepared as a TFA salt. 5 mg/kg is a similar, but slightly lower dose than what has been used in mice for the parent compound of SNS-032 for cancer (xenograft model, 15 mg/kg[1]). We wanted to see if we could achieve efficacy at a slightly lower dose with our compound.

How were the doses used for the brain/plasma partitioning chosen?

We selected the same dose as was used for the Snapshot PK (2.29 mg/kg) to be able to have a more comprehensive profile of the compound at that dose. Since brain penetration was anticipated to be low, a higher concentration was also elected. Given the potency of B1 in vitro and to avoid potential toxicities caused by inhibition of off-targets, we chose to use lower concentrations than have been reported for SNS-032 (15 mg/kg[1]). Our goal is not to kill cancer cells, so we would like to find an efficacious dose without associated toxicity. Some of these details have been noted in the Analysis of Brain/Plasma Concentration Methods section.

Details on the housing and feeding of the laboratory animals at The Francis Crick Institute should be provided.

We added additional information to the Methods: “Animals were housed in a controlled environment with 12-hour light/dark cycle. They were fed ad libitum and used in accordance with the Animals (Scientific Procedures) Act 1986 of the United Kingdom. Protocols and procedures were approved by The Francis Crick Institute project license P5E6B5A4B."

What is the total number of compounds included in the extensive libraries of SNS-032 and AT-7519 analogs?

This detail has been added to the Results section. We profiled our extensive library of more than 50 SNS-032 and more than 100 AT-7519 analogs using the CDKL5 NanoBRET assay.

Whereas the S10 score is defined in the Figure 3 caption, it would be a good idea to include its definition in the main text, possibly in the Method section.

Thanks for this suggestion. The S_10_ score is now defined in the Methods section as well, with details on how to calculate it.

Have the authors obtained data on the unbound fraction of B1?

No, not at this time.

I am not convinced by the argument that, since B1 reduces the phosphorylation of the specific substrate EB2, interference with other kinases expressed in the brain might not be at play here. Please elaborate further.

We have rewritten this section to include additional data supporting that at the concentrations used (≤100 nM) that only CDKL5 is efficiently engaged by B1 in vitro. While this does not exclude the possibility that other kinases not included in the selectivity profiling are involved, the “unscreenable” kinome or other ATP competitive proteins would be required to bind B1 at ≤100 nM in order to influence the biology observed. While possible since we are screening ~80% of the kinome, this seems less likely.

The authors indicate that brain penetration of B1 is rather low. Do they see this as an obstacle to using their inhibitor to further study the role of CDKL5 in CDD using in vivo assays? What about determining the kpuu, which is much more relevant pharmacologically speaking? (the shift in potency from biochemical to cell-based assays also suggests a permeability problem). Haven´t the authors considered a hit-to-lead campaign to improve B1 PK profile?

In the Discussion we have further qualified the use of B1 as a cellular tool to study the pathophysiology associated with CDKL5 dysfunction in vitro.

There are other processes outside the brain (ie, acute kidney injury), where CDKL5 inhibition via an inhibitor that does not cross the BBB would be of greater utility[2, 3]. Future in vivo studies using B1 via implantation of minipumps that are capable of stable dosing (and inhibition) over extended periods of time are still possible. In this instance, the effect of CDKL5 inhibition on behavior and seizure susceptibility could be directly studied. Furthermore, we have added text to indicate that a hit-to-lead campaign could furnish an analog better suited for in vivo studies. A hit-to-lead campaign would benefit from determining the kpuu for leads, as the reviewer suggests. The IC_50_ of B1 in cell-free assays (Luceome KinaseSeeker and CDKL5 enzymatic assays) is nearly equivalent to its IC_50_ in the cell-based CDKL5 NanoBRET and Western blot assays: all between 1–10 nM. This supports that there is no shift in potency from biochemical to cell-based assays or permeability problems.

The potential utility of B1 could be further highlighted towards the concluding section of the manuscript.

We have highlighted in the last paragraph of the Discussion that B1 is one of the best available chemical probes for interrogating CDKL5 activity in cells and brain slices, especially across development.

I suggest indicating the statistically significant differences in the means in Figure 6 graphically. Can the authors explain the apparent lack of a clear dose-response behavior?

We have added this to Figure 6. While a dose-response relationship is clear with cultured neurons (Figure 5), it is not readily apparent in brain slices as presented in Figure 6. There are possible reasons. Both 45nM and 100 nM are highly effective in bringing pEB2 close to background levels, thus comparison between these doses may be problematic. Also, monolayer cultures have immediate access to inhibitors which may result in lower variability.

Seemingly, the authors have not observed their code to indicate different levels of statistical significance in Figure 7.

Figure 7 has been adjusted for clarity.

What conclusion(s) can the authors draw from the similar microsomal stability observed with and without NADPH? Are the differences between 0 and 30 min statistically significant? Please provide a wider interpretation of the results.

Additional relevance of running the assay in the presence and absence of NADPH has been added to the Methods section. The Results section was supplemented with some additional clarification that non-NADPH dependent enzymatic degradation was not observed when B1 was analyzed. The differences between 0 and 30 min are statistically significant based on what we have observed in analyzing other published and unpublished kinase inhibitory scaffolds that significantly degrade after just 30 minutes. For example, after 30 min incubation with mouse liver microsomes, only 12.6% of a PIKfyve chemical probe and 40% of a CK2 chemical probe remained. We have added a relevant reference to support this in the Results section.

Reviewer #2 (Recommendations for the authors):I have several questions to clarify some descriptions.1. Selectivity scoreFigure 3 and the corresponding description of the selectivity score – the meaning of these is hard to understand. Do the dots in kinome tree diagrams represent 403 kinases? Which dot is CDKL5? Why were red circles drawn only on the CMGC family? The authors described that the S10 score is a way to express selectivity that corresponds with the percent of the kinases screened that bind with a PoC value <10 and indicated 0.017, 0.02, and 0.01 for B1, B4, and B12 respectively. But it is hard to interpret the significance of these values. Can you provide some scale or perspective and let us know how selective these values are in terms of the kinome? Why do some kinases have two IC50 (nM) values?

Thank you for drawing our attention to this. We have supplemented the Methods section that describes the DiscoverX scanMAX panel with details about how selectivity score is calculated and what it means. Some additional perspective and examples were added to further qualify the selectivity scores. To further address the concerns raised, we have also clarified in the legend that the dots represent kinases that bind with high affinity to the compound being assayed. Furthermore, we have labeled all red dots with the corresponding kinase. We have added a sentence to the discussion to highlight the important point about these compounds retaining CMGC family potency. Finally, the tables in Figure 3 have been reconfigured to have separate columns for biochemical (binding or enzymatic) and NanoBRET data. The legend has been updated to reflect this change as well.

2. Figure 7BPlease add legends for 3 traces.

Figure 7B and the legend has been modified for clarity.

3. Lines 653-656"At issue, is whether observations of CDKL5 activity, either through genetic manipulations or interference with the kinase, are specific and not potentially subsumed by the activity of other kinases. Given the reduction of phosphorylation of the specific substrate, EB2, this is less likely here."It is hard to understand the intent of these sentences.

We agree with the reviewer that this is not clear. This text has been deleted. The more comprehensive NanoBRET profiling of CAF-382 (B1) supports that at the concentrations we used that the results generated are specific to CDKL5 inhibition and not due to inhibition of offtarget kinases.

Reviewer #3 (Recommendations for the authors):The first part of the paper describing binding and orthogonal assays should be improved to render the read easier. For non-expert readers, it is impossible to really understand the specificity of the inhibitor, which is a central part of the manuscript. It is also not clear why the authors start with a panel of 25 inhibitors and test 20 in the rat neurons.

We have added a lot more detail to the descriptions of the orthogonal assays in the Methods and Results sections. This provides more context about what each assay is meant to evaluate and how to interpret the results. We tested all 25 putative CDKL5 lead compounds (Table S1) in rat neurons (Figure 1).

While the work in general is well presented it could be improved by being updated and, in a few cases, more precise.Line 170: "epilepsy in CDD does not respond to any seizure medication" – that is a bit overstated.

We have cited the literature that supports this statement. While I have not come across such an individual (TAB, I have personally seen and examined over 100 individuals with CDD) and with our current clinical studies involving CDD (NIH/NINDS U01), all patients (>150) meet the ILAE definition of medically refractory (https://www.ilae.org/patient-care/drug-resistant-epilepsy), these have been reported so we have adjusted this statement and added those references.

Line 182: I suggest citing Terzic and colleagues' work from 2021 showing that CDKL5 is required also during adulthood.

Thank you. The reference to Terzic et al. have now been cited in this section.

Line 192-197: I suggest including De Rosa et al's work from 2023 suggesting a role of CDKL5 in GABAA-R mediated inhibition through the interaction with gephyrin.

Thank you. This is an important reference that has now been added.

Line 590: the work by Kontaxi et al. has been published.

The reference has been updated.

Kinase assay: it is not clear whether full-length CDKL5, and which isoform, was used for the in vitro kinase assay. Furthermore, it is not clear why myelin basic protein was used as a substrate – does it contain the CDKL5 consensus site (which is by the way never mentioned in the paper)?

We have added that we used full-length human CDKL5 (transcript variant II, NCBI Reference Sequence: NM_001037343.2) to the Methods section. We added to the Methods that isoforms are homologous through the kinase domain.

The basis for our selection of myelin basic protein as a substrate has been explained with references to support this choice. To execute the in vitro kinase assays, myelin basic protein (Active Motif, 31314) was employed as a substrate for recombinant CDKL5. Myelin basic protein is used as a substrate for multiple kinases, both serine/threonine and tyrosine kinases, to enable in vitro kinase assays due to the presence of multiple sites for phosphorylation. As such, we and others have used this protein as a kinase substrate for evaluating kinase activity[2, 4]. MBP does not contain a CDKL5 consensus site of RPXS/T*, and as such could be considered a less than ideal substrates to study CDKL5 activity, however for in vitro kinase assays MBP is still suitable as it can be phosphorylated by CDKL5. In addition, CDKL5 is known to phosphorylate substrates that do not contain a consensus motif[3].

References

Wu, Y., et al., Cyclin-dependent kinase 7/9 inhibitor SNS-032 abrogates FIP1-like-1 platelet-derived growth factor receptor α and bcr-abl oncogene addiction in malignant hematologic cells. Clin Cancer Res, 2012. 18(7): p. 1966-78.

Kim, J.Y., et al., A kinome-wide screen identifies a CDKL5-*SOX9* regulatory axis in epithelial cell death and kidney injury. Nat Commun, 2020. 11(1): p. 1924.

Kim, J.Y., et al., Involvement of the CDKL5-*SOX9* signaling axis in rhabdomyolysis associated acute kidney injury. Am J Physiol Renal Physiol, 2020. 319(5): p. F920-F929.

Kameshita, I., et al., TandeMBP: generation of a unique protein substrate for protein kinase assays. J Biochem, 2014. 156(3): p. 147-54.